



# Can seasonal hydrological forecasts inform local decisions and actions? A decision-making activity

**Jessica L. Neumann**[1], **Louise Arnal**[1,2], **Rebecca E. Emerton**[1,2], **Helen Griffith**[1], **Stuart Hyslop**[3], **Sofia Theofanidi**[1], and **Hannah L. Cloke**[1,4,5]

[1]Department of Geography and Environmental Science, University of Reading, Reading, UK
[2]European Centre for Medium-Range Weather Forecasts (ECMWF), Reading, UK
[3]Environment Agency, Kings Meadow House, Reading, UK
[4]Department of Meteorology, University of Reading, Reading, UK
[5]Department of Earth Sciences, Uppsala, Sweden

**Correspondence:** Jessica L. Neumann (j.l.neumann@reading.ac.uk)

**Abstract.** While this paper has a hydrological focus (a glossary of terms highlighted by asterisks in the text is included in Appendix A), the concept of our decision-making activity will be of wider interest and applicable to those involved in all aspects of geoscience communication.

Seasonal hydrological forecasts (SHF) provide insight into the river and groundwater levels that might be expected over the coming months. This is valuable for informing future flood or drought risk and water availability, yet studies investigating how SHF are used for decision-making are limited. Our activity was designed to capture how different water sector users, broadly flood and drought forecasters, water resource managers, and groundwater hydrologists, interpret and act on SHF to inform decisions in the West Thames, UK. Using a combination of operational and hypothetical forecasts, participants were provided with three sets of progressively confident and locally tailored SHF for a flood event in 3 months' time. Participants played with their "day-job" hat on and were not informed whether the SHF represented a flood, drought, or business-as-usual scenario. Participants increased their decision/action choice in response to more confident and locally tailored forecasts. Forecasters and groundwater hydrologists were most likely to request further information about the situation, inform other organizations, and implement actions for preparedness. Water resource managers more consistently adopted a "watch and wait" approach. Local knowledge, risk appetite, and experience of previous flood events were important for informing decisions. Discussions highlighted that forecast uncertainty does not necessarily pose a barrier to use, but SHF need to be presented at a finer spatial resolution to aid local decision-making. SHF information that is visualized using combinations of maps, text, hydrographs, and tables is beneficial for interpretation, and better communication of SHF that are tailored to different user groups is needed. Decision-making activities are a great way of creating realistic scenarios that participants can identify with whilst allowing the activity creators to observe different thought processes. In this case, participants stated that the activity complemented their everyday work, introduced them to ongoing scientific developments, and enhanced their understanding of how different organizations are engaging with and using SHF to aid decision-making across the West Thames.

## 1 Introduction

There has been a recent shift away from the conventional linear model of science, where research is carried out within the scientific community with the expectation that users will be able to access and apply the information, towards co-production and stakeholder-led initiatives that bring together scientists and decision-makers to frame and deliver "actionable research" (Asrar et al., 2012; Lemos et al., 2012; Meadow et al., 2015). Regular and clear communication between scientists and policy-makers and practitioners in

workshops, focus groups, consultations, and interviews, and through the development of games, activities, and interactive media, is imperative for ensuring that projects deliver impact outside of the academic environment. Here, we share findings from an activity that explored the use of seasonal hydrological forecasts* for local decision-making. This was conducted as part of an IMPREX (IMproving PRedictions and management of hydrological Extremes) stakeholder focus group for the West Thames, UK (van den Hurk et al., 2016; IMPREX, 2018a), co-organized by the University of Reading (UoR), UK, Environment Agency (EA) and supported by the European Centre for Medium-Range Weather Forecasts (ECMWF).

Seasonal hydrological forecasts (SHF) have the ability to predict principal changes in the hydrological environment such as river flows and groundwater levels weeks or months in advance. This has the potential to benefit humanitarian action and economic decision-making, e.g. to provide early warning of potential flood and drought events, assist with water quality monitoring, and ensure optimal management and use of water resources for public water supply, agriculture, and industry (Chiew et al., 2003; Arnal et al., 2017; Li et al., 2017; Meißner et al., 2017; Turner et al., 2017). SHF systems covering a range of spatial scales have been developed – Hydrological Outlook UK forecasts at a national level (Prudhomme et al., 2017; CEH, 2018) – while the Copernicus European and Global Flood Awareness Systems (EFAS and GloFAS) provide operational forecasts over larger scales (JRC, 2018a, b). Recent research has demonstrated improvements in SHF quality*, including increased accuracy out to 4 months for high-flow events during the winter in Europe (Arnal et al., 2018; Emerton et al., 2018).

There is growing interest in SHF amongst policy-makers and practitioners; however, in many cases, there is limited information about whether SHF products are *actually* being used. Research output has focused largely on technical system development and improvements to forecast skill* (see the review by Yuan et al., 2015), with relatively fewer studies exploring how users engage with and apply SHF to inform decisions (see Crochemore et al., 2015; Viel et al., 2016). Many seasonal forecasting studies, including those investigating the application of seasonal meteorological forecasts* (which provide information about future weather variables, rather than hydrology more specifically), have identified forecast uncertainty*, whereby forecast skill and sharpness* decrease with increasing lead time* (Wood and Lettenmaier, 2008; Soares and Dessai, 2015), and how this uncertainty can be communicated effectively as key barriers to use (Arnal et al., 2016; Vaughan et al., 2016). Non-technical factors, including the level of knowledge and training required to interpret and apply SHF information effectively (Bolson et al., 2013; Soares and Dessai, 2016), the visualization, format, and compatibility of the information provided (Fry et al., 2017; Soares et al., 2018), and the level of communication between different users in the water sector and between

research developers and practitioners (Golding et al., 2017), have all been found to act as both barriers and enablers, depending on the user group in question.

The potential for SHF to meet the needs of the water sector is recognized by a host of UK environmental organizations, including the EA, the Met CE1 Office, and research centres (see Prudhomme et al., 2017). The West Thames specifically is underlain by a slowly responding, largely groundwater-driven hydrogeological system (Mackay et al., 2015), meaning that there is potential for extreme hydrological events such as the drought of 2010–2012 (Bell et al., 2013) and winter floods of 2013–2014 (Neumann et al., 2018) to be detected weeks or months in advance. It also has a dense population and high demands for water which require effective long-term management of resources for public drinking supply, industry, agriculture, and wastewater treatment (further details about the West Thames can be found in Sect. 2.2). The value of using SHF in the West Thames is of particular interest to the EA; however, information on the level of understanding, uptake, and application is currently unknown. We therefore aimed to develop a clearer understanding about how different professional water sector users – broadly forecasters, groundwater hydrologists, and water resource managers – are currently engaging with SHF in the West Thames using a decision-making activity.

In the context of flood science communication with experts, real-time activities such as simulation exercises (that imitate real-world processes and behaviours) or roleplay (where participants engage with real-world scenarios but take on personas and positionalities that differ from their own) are known to be effective when engaging with stakeholders who bring a range of scientific ideas and perspectives to the table (McEwen et al., 2014). Such activities encourage participants to apply their knowledge to realistic situations and to reflect on issues and the perspectives of other stakeholders (Pavey and Donoghue, 2003, p. 7). They are also valuable for understanding decision-making processes, e.g. for environmental hazards and conflicting community views (Harrison, 2002), for capacity building in response to new water legislation (Farolfi et al., 2004), and for understanding climate forecasts and decision-making (Ishikawa et al., 2011). Our decision-making activity provided an interactive and entertaining platform that encouraged participants to engage with real-world scenarios whilst fostering discussions about the barriers and enablers to use of SHF. Using three activity stages, participants were provided with sets of progressively confident and locally tailored SHF for the next 3 to 4 months. The SHF were produced using output from operational systems including Hydrological Outlook UK and the European Flood Awareness System (EFAS), and hypothetical forecasts generated through scientific research (see Neumann et al., 2018). Participants were asked to play in real time, i.e. as if receiving the forecasts on the day for the next 3 to 4 months. They did not know in advance whether the SHF represented a flood, drought, or business-as-usual scenario

and had to use their knowledge and experiences to make informed decisions based on the maps, hydrographs*, tables, and text provided. In reality, all three sets of SHF represented the same time period: winter 2013–2014 (a period of extensive flooding nationwide that occurred at the end of 2 years of drought conditions in the UK). Between December 2013 and February 2014 the West Thames experienced extreme flooding from fluvial and groundwater sources which had knock-on impacts for local water quality, sewage treatment, and water resource management – opening up discussions for all participants. Given that issues relating to flood and drought risk, water quality, and water resource management in the West Thames are generally managed by local and regional-area authorities (Thames Water, 2010), the activity focused on whether SHF can be used to support decision-making at the local level. To the best of our knowledge, this scale of practical application has yet to be explored, we suspect mainly due to the lower skill of seasonal meteorological forecasts in Europe, particularly with respect to precipitation, which is a key variable of interest for hydrology (Arribas et al., 2010; Doblas-Reyes et al., 2013). A brief overview of the focus group is provided in Sect. 2, the full activity set-up is detailed in Sect. 3, and the findings and the discussion are presented in Sects. 4 and 5.

## 2 Overview of the focus group

### 2.1 Aims of the focus group

The focus group was developed in collaboration with the EA and in line with the objectives of the IMPREX project. The aims were the following.

- Introduce and discuss current SHF projects, products, and initiatives for the UK and Europe.

- Engage with participants' experiences and knowledge of using SHF.

- Learn how SHF are being applied in the West Thames and recognize how different users in the water sector approach and apply SHF information for decision-making.

- Identify limitations and barriers to use.

- Identify future opportunities for SHF application and research.

These aims were delivered through a series of four interactive sessions designed to actively engage participants to share their knowledge and experiences of SHF, and short presentations that introduced the main topics surrounding SHF and informed participants about current SHF projects and developments in the scientific research. While this paper focuses on the decision-making activity (interactive session 2), discussions from the other sessions are also presented where

relevant. An outline of the focus group programme is provided in Supplement 1 and a full report of the activities is available; see Neumann et al. (2017).

### 2.2 The West Thames in southern England

#### 2.2.1 Physical geography

The West Thames refers to the non-tidal portion of the Thames River Basin*, from its source in the Cotswolds in the west of England to 230 km downstream at Teddington Lock in western London (Fig. 1). It covers an area of 9857 km$^2$ TS1 (the Thames basin is 16 980 km$^2$) and comprises 10 river catchments* that are the tributaries* that feed directly into the River Thames (Fig. 1). The western catchments are predominantly rural; land use is a mix of agriculture and woodland with rolling hills and wide, flat floodplains (elevation up to 350 m a.s.l.). Towards the centre and east, the region becomes increasingly urbanized, encompassing the towns of Reading and Slough and outskirts of Greater London (elevation 4 m a.s.l. at Teddington Lock). Lithology* varies markedly across the West Thames. Catchments overlaying the Cotswolds (upstream) and the Chilterns (middle sections) are dominated by chalk and limestone aquifers* with high baseflow*, while a band of less-permeable clays and mudstones separates these two areas. Sandstones, mudstones, and clays are also prevalent towards London (downstream) – these catchments have higher levels of surface runoff* and can exhibit a flashier* response to storm events (Bloomfield et al., 2011; EA, 2009).

#### 2.2.2 Water demands, risk, and management – why the West Thames is of interest

The West Thames is a highly pressured environment – 15 million people and a substantial part of the UK's economy rely directly on its water supply (EA, 2015). There are more than 2000 licensed abstraction points in the chalk aquifers and superficial alluvium and river terrace gravel deposits; 90 % of abstractions are for public water supply, the rest providing water for agriculture, aquaculture, and industry (Thames Water, 2010). There are 12 000 registered wastewater discharge points; pollution from sewage treatment works, transport, and urban areas affects more than 45 % of rivers, water bodies, and aquifers, largely towards London. Diffuse pollution and sedimentation from agricultural and forestry practice are the main contributors to poor water quality in the upper catchments, especially during times of high rainfall (EA, 2015).

Urbanization and land-use change in combination with more varied rainfall patterns have seen the region affected by a number of extreme drought and flood events in recent years (EA, 2009; Parry et al., 2015; Muchan et al., 2015). Across the Thames Basin, 200 000 properties are at risk from a 1 : 100*-year fluvial flood, with 10 000 at risk from a 1 : 5*-year event (EA, 2009). Low and high river flows also pose

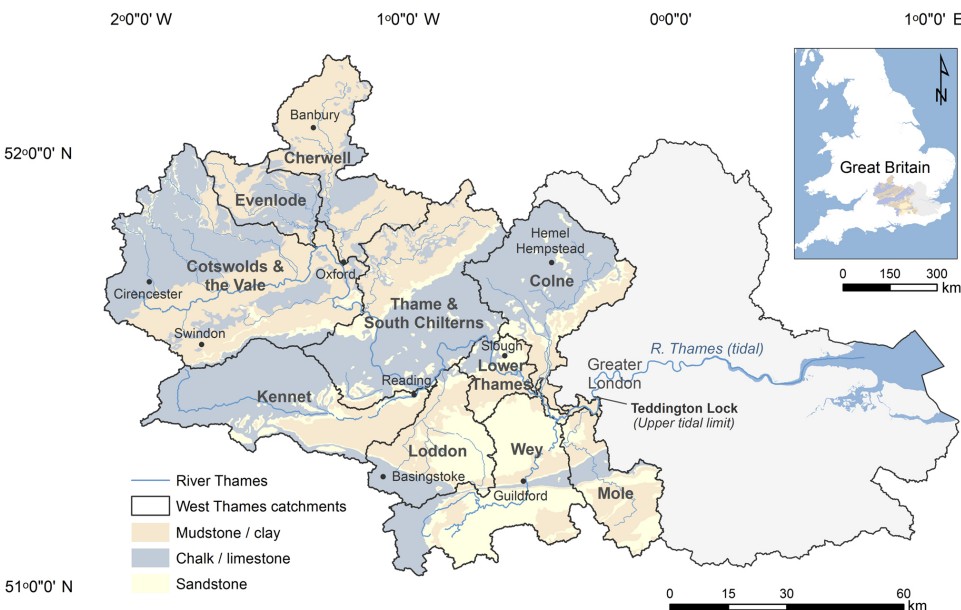

**Figure 1.** Location and lithology of the West Thames and its 10 main river catchments.

risks to navigation and management of the canal network which is highly important for recreation, local living, and the economy (Wells and Davis, 2016).

## 2.3 Participants

### 2.3.1 Who took part?

SHF have the potential for wide-ranging application and it was important to capture the different perspectives of the West Thames water sector. The organizers agreed that the focus group would work well with a relatively small number of participants (up to 12) so that all perspectives could be heard. Based on discussions held between the organizers, individuals from local organizations working in established (i.e. long-term/permanent/leadership) roles relevant to SHF in the West Thames were invited; many but not all participants had previously collaborated with the University of Reading and/or EA. In some cases, an invitee was unable to attend due to prior commitments or because they had a colleague who they felt would be a better fit for the focus group. A total of 17 participants were invited from six organizations – 12 accepted and 11 took part on the day. They were responsible for flood and drought forecasting (F × 3), groundwater modelling and hydrogeology (GH × 2), navigation (N × 1), water resource and reservoir management (WR × 2), public water supply (WS × 2), and wastewater modelling and operations (WW × 1). They represented five organizations: two non-departmental public bodies (sponsored by government agencies), two science and research centres, one water service company, and one non-for-profit organization (Table 1).

### 2.3.2 Current engagement with SHF

By inviting local stakeholders we ensured that participants represented a range of different water sector personas and were familiar with the West Thames environment. We did not assume that participants had any prior knowledge of SHF and invitees were encouraged to attend even if they were unfamiliar with the concept as this would be an important indicator of the state of play in the West Thames (invite poster; see Supplement 1).

All 11 focus group participants were familiar with the concept of seasonal hydrological forecasting and 10 regularly used SHF in their everyday job (according to results from interactive session 1 – "What are seasonal hydrological forecasts?"). Using post-its, participants noted that Hydrological Outlook UK (CEH, 2018) and the associated raw forecasts from the analogue, hydrological, and meteorological models (produced by the UK Met Office, Centre for Ecology and Hydrology, British Geological Survey, EA, Natural Resources Wales, Scottish Environment Protection Agency, and Rivers Agency Northern Ireland) were the main sources of SHF information currently being used, primarily for flood and drought outlook, groundwater monitoring, and river flow projection purposes. Scientific research, operational planning, and sharing of information with other organizations in the water sector were also listed as reasons for engaging with SHF. It is important to note that no prior definitions or information were provided and no restrictions or guidance were placed on what participants should write down. This suggests that many in the water sector are using SHF to obtain an insight into whether the upcoming season will be drier or wetter than normal, but that they also believe SHF *potentially*

**Table 1.** Breakdown of participants who took part in the activity.

| Job title | Organization type | Role in the activity |
|---|---|---|
| Modelling and Forecasting Team Leader | Public body/government agency (1) | Flood and drought forecaster |
| Chief Hydrometeorologist | Public body/government agency (2) | Flood and drought forecaster |
| Climate Scientist (Professor) | Science and research centre (1) | Flood and drought forecaster |
| Thames Water Resources Technical Specialist | Public body/government agency (1) | Groundwater modelling and hydrogeology |
| Groundwater Research Directorate | Science and research centre (2) | Groundwater modelling and hydrogeology |
| Principal Hydrologist for Water Management | Not-for-profit (charitable trust) | Navigation |
| Water Resources, Environment and Business Directorate | Public body/government agency (1) | Water resource and reservoir management |
| Abstraction and Transfers Analyst | Water service company | Water resource and reservoir management |
| Water Strategy and Resources Modeller | Water service company | Public water supply |
| Thames Region Hydrologist | Public body/government agency (1) | Public water supply |
| Wastewater Modelling Specialist | Water service company | Wastewater modelling and operations |

have the capability to forecast possible flood and drought risk, which could be used to support decision-making and provide better preparedness. This is an encouraging starting point, although many participants noted that this potential is not currently being realized due to the uncertainty and coarse spatio-temporal resolution of SHF; e.g. Hydrological Outlook UK forecasts are only published monthly for the main UK river basins.

## 3  Set-up of the decision-making activity

### 3.1  Background

Our activity was inspired by the success of previous decision-making activities and games run by the HEPEX (Hydrological Ensemble Prediction EXperiment) community (e.g. Ramos et al., 2013; Crochemore et al., 2015; Arnal et al., 2016). The aim was to better understand how different water sector users in the West Thames interpret and act on SHF by providing them with hydrological context, maps, and forecasts for the region. The activity was designed for the West Thames so that we could capture the relationship between local stakeholders and the environment in which they work.

### 3.2  Activity design

#### 3.2.1  Overview of the set-up

The set-up of the activity (illustrated in Fig. 2) had the following structure: Choose groups > Define the Objectives > Background Context > Stage 1 > Stage 2 > Stage 3. Participants divided themselves into three groups based on their area of expertise and where they felt they could best contribute to the discussions. There were three flood and drought "forecasters" and two "groundwater hydrologists". The remaining participants (navigation, water resource and reservoir management, public water supply and wastewater operations) grouped themselves as "water resource managers". While the results and discussions focus on these three broad groups, individual perspectives are also included to

capture the variety of water sector personas present. There were also three research facilitators and three note-takers whose role it was to capture and record the key discussion points.

Groups were first provided with background context to the West Thames to set the scene, followed by three sets of progressively confident SHF for the next 3 to 4 months (Stages 1–3). Stage 1 forecasts were from Hydrological Outlook UK, Stage 2 were from EFAS-Seasonal (European Flood Awareness System) and Stage 3 were "improved" output from EFAS-Seasonal (Fig. 2 and Sect. 3.4). Participants were asked to discuss the information presented in their groups and make informed decisions about each of the 10 West Thames catchments (Fig. 1 and Sect. 3.3.2). All groups were provided with exactly the same information and discussion was encouraged. The activity took around 2 h and timings were only loosely controlled.

SHF at all three stages of the activity represented the same time period – dating from 1 November 2013 to 28 February 2014 (or 31 January 2014 for Hydrological Outlook UK, which only extends to 3 months; CEH, 2018). These dates captured a period of severe and widespread river and groundwater flooding in the West Thames (Huntingford et al., 2014; Kendon and McCarthy, 2015; Muchan et al., 2015). *Participants did not know the dates of the forecasts, nor were they informed whether the situation being forecasted was a high flow (flood), low flow (drought) or a business-as-usual scenario.* Dates were removed from all information, and streamflow- and groundwater-level units were removed from the Stage 2 and Stage 3 EFAS hydrographs, although exceedance thresholds were provided for context. The decision to remove units was advised by the EA. The concern was that participants familiar with average and high-flow values for specific catchments would deduce that the SHF must represent the 2013–2014 floods, which would bias their decision-making based on their previous experience and memories. No information on forecast skill or quality was given and participants were asked to treat all information as

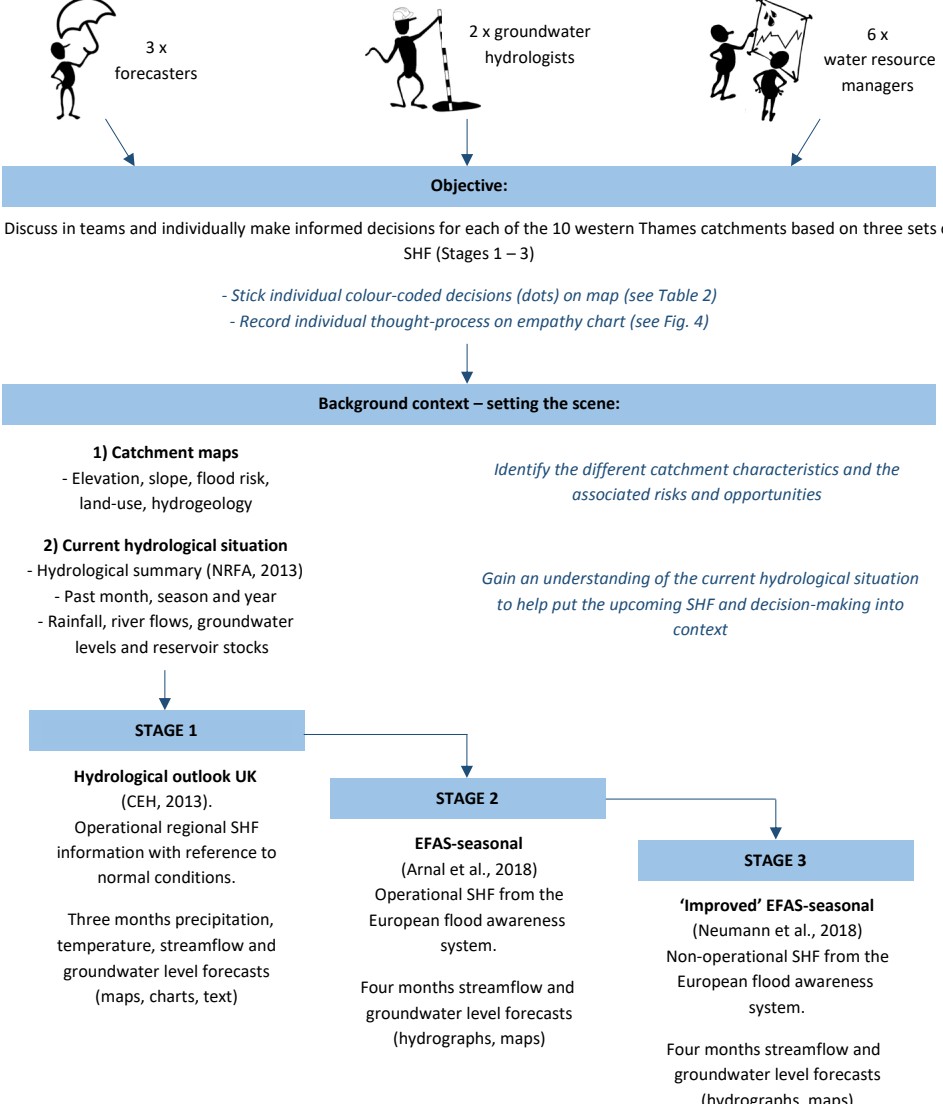

**Figure 2.** Set-up of the activity.

being "current", i.e. as if receiving the SHF today, for the next 3–4 months to create a realistic forecasting scenario.

### 3.2.2  Recording the decisions

In real life, a user's decision process can encompass a range of possible actions and associated consequences (Crochemore et al., 2015). Decisions can be controlled by providing participants with a set of options to choose from, e.g. to deploy temporary flood defences or not – the consequences of which usually determine the outcome of a game or activity. In this case, participants were asked to select from a broad range of colour-coded options (Table 2), but specific decisions were not defined as these had the potential to differ greatly between participants and might prompt unrealistic answers. At each stage, the colour-coded options were discussed by the three groups, simulating conversations that could happen in real life, but it was stressed that *the colour chosen was to be representative of what an individual participant, or their organization, would do with the SHF information in each catchment*. This was recorded on an A1 map using coloured sticky dots marked with the participant's initials ($n \sim 110$ dots per map (11 participants, 10 catchments)) (Fig. 3). In cases where participants were not familiar with all catchments, or did not feel able to make an informed decision, they did not place a dot. It was important to gather a written record explaining how and why the decisions were reached, and so participants were also asked to complete an A4 empathy map at each stage (Fig. 4). Originally designed as a collaborative tool to be used in business and marketing,

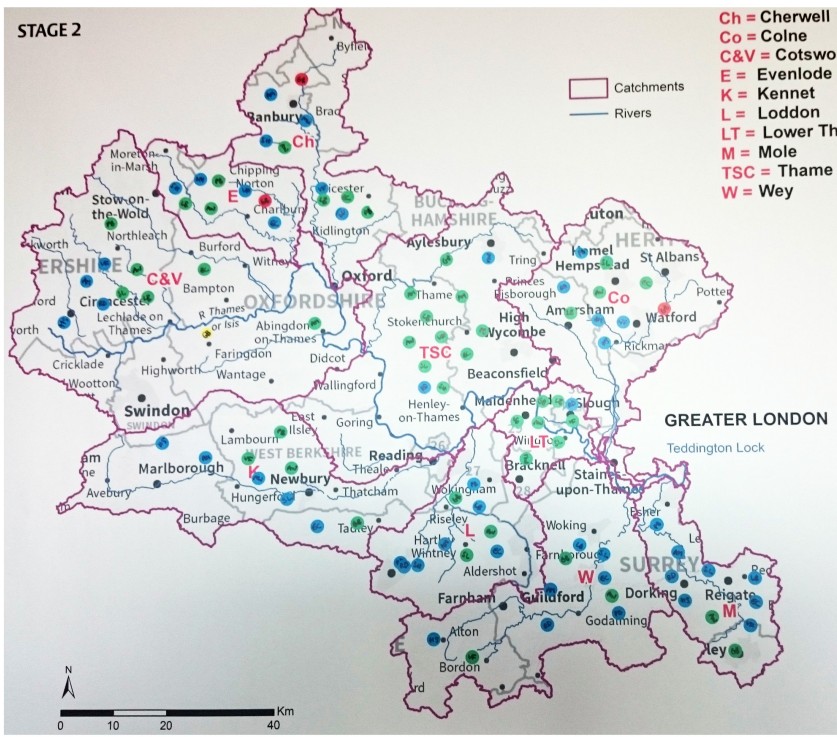

**Figure 3.** Participants' individual colour-coded decisions recorded on an A1 map.

**Table 2.** Colour codes and corresponding action or decision to be taken.

| Decision to be made or action to be taken |
| --- |
| ● Ignore the SHF information: wait for the more skilful forecasts with shorter lead times (e.g. a 7–10-day forecast). |
| ● Look at the SHF information: decide there is no notable risk and do nothing at this point. |
| ● Look at the SHF information: discuss or pass the information on to relevant colleagues/departments in your organization and agree to keep an eye on the situation. |
| ● Look at the SHF information: discuss or pass the information on to relevant colleagues/departments in your organization *but also* external partners – actively request further information about the situation or seek advice on possible actions. |
| ● Look at the SHF information: decide to implement or set in motion action(s) in a catchment, e.g. to help with drought preparedness, early warning, repairs, or maintenance to flood defences. |

empathy maps aim to gain a deeper understanding about an external user's experiences and decisions (Gray, 2017). Here, we adapted the traditional use by asking individuals to reflect on their own decisions based on their real-life experiences and discussions with other group members. This allowed us to capture individuals' thought processes, influences, discussions, and the potential risks and gains associated with their decision (Fig. 4). By combining the information recorded on empathy maps for each group, we also gathered an overview of the shared understanding between forecasters, groundwater hydrologists, and water resource managers and how their SHF needs and expectations match and differ when it comes to decision-making.

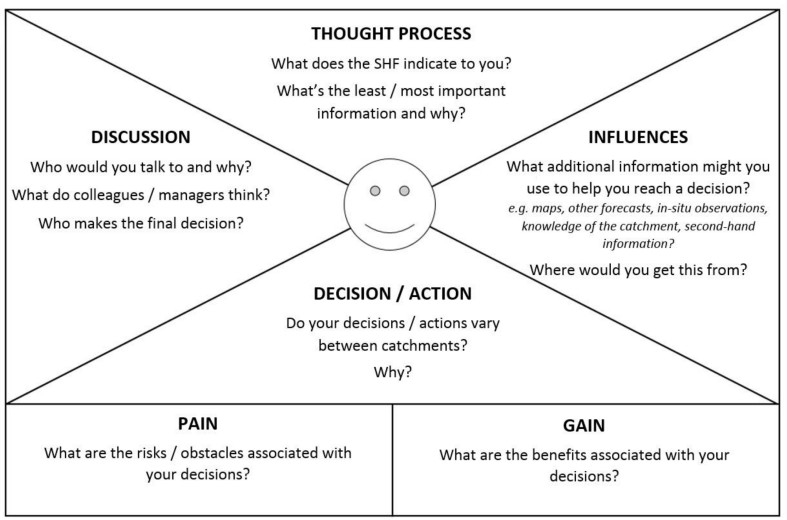

**Figure 4.** Empathy map completed by each participant during Stages 1–3.

## 3.3 Background context

Groups were given information about the West Thames catchment characteristics and "current" hydrological conditions (units and dates removed) to place the upcoming SHF into context and aid interpretation.

### 3.3.1 Catchment characteristics – driving factors, risks and opportunities

Five maps (Supplement 2) that provided a visual representation and a numerical breakdown of the characteristic differences between each catchment were given to participants.

- Hydrogeology* – dominant geological type (sandstone, chalk, clay)

- Elevation – minimum, maximum and mean elevation (m a.s.l.)

- Slope – minimum, maximum and standard deviation of slope angle (degrees)

- Land cover – dominant land use (urban, woodland, agricultural, semi-natural)

- Flood risk – flood warning and flood alert areas and an indication of "urban flood risk"

Participants were asked to discuss and identify the key differences between catchments and highlight the associated risks and opportunities. As some participants were more familiar with specific areas/catchments based on their day job, the maps provided a wider view of where catchment characteristics differ across the West Thames region.

### 3.3.2 Current hydrological situation

To help set the scene with respect to initial conditions, i.e. the "current" levels of water contained in the soil, groundwater, rivers, and reservoirs, groups were provided with information from the Hydrological Summary (NRFA, 2018TS2) for the last month, past season, and past year (October 2013, June to September 2013, and November 2012 to October 2013 with dates removed). The Hydrological Summary (Supplement 3) focuses on rainfall, river flows, groundwater levels, and reservoir stocks and places the events of each month, and the conditions at the end of the month, into a historical context. In the real world, decision-makers are already prepared with this information; thus, providing evidence about whether hydrological conditions were wet, dry, or normal at the point of receiving the forecasts was an important piece of information for the participants to consider.

## 3.4 Activity Stages 1–3: the seasonal hydrological forecasts

### 3.4.1 Stage 1 – Hydrological Outlook UK

The first set of SHF information provided to participants was the Hydrological Outlook UK (from 1 November 2013 to 31 January 2014, with dates removed) (CEH, 2013). This provided regional information for the next 3 months with reference to normal conditions for precipitation, temperature, river flows and groundwater levels. Hydrological Outlook UK uses observations, ensemble models and expert judgement (CEH, 2018) to produce the seasonal forecasts. Information is publicly available and consists of text, graphs, tables and regional maps (examples are shown in Fig. 5 and the full set of forecasts provided to participants are in Supplement 4).

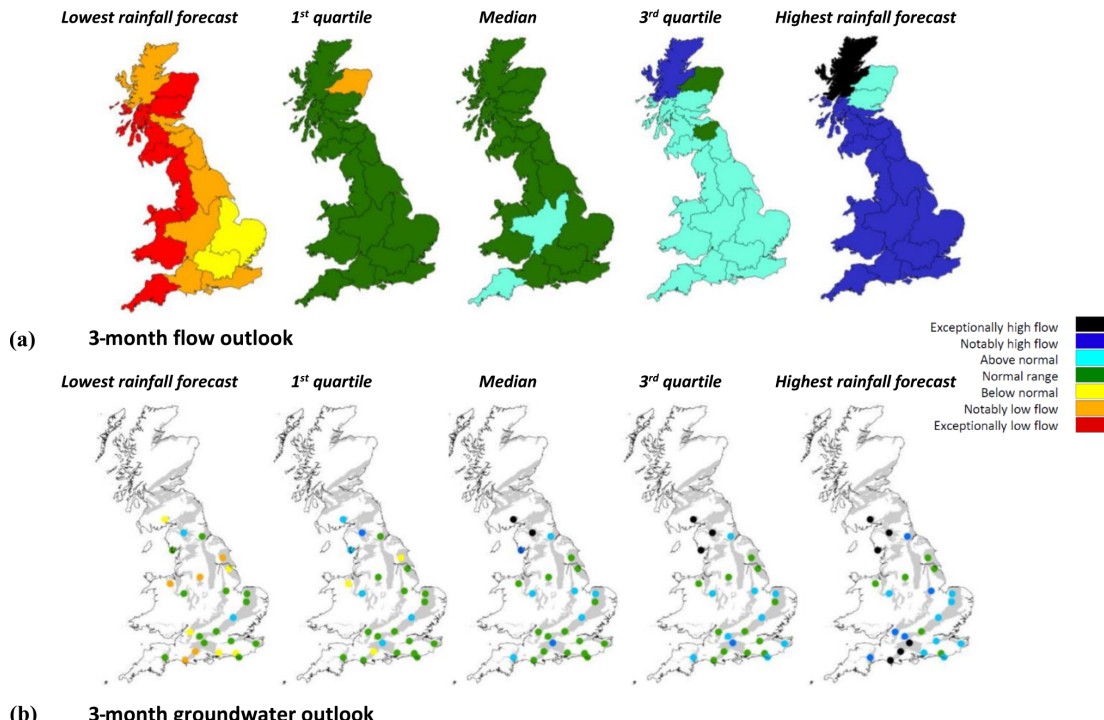

**Figure 5.** UK 3-month outlook maps from November 2013 (colours based on the percentile range of historical observed values). **(a)** Regional river flow forecasts created from climate forecasts. **(b)** Groundwater level forecasts at 25 UK boreholes created from climate forecasts (CEH, 2013).

### 3.4.2 Stage 2 – EFAS-Seasonal

EFAS-Seasonal (European Flood Awareness System) is an operational system that monitors and forecasts streamflow* across Europe, with the potential to predict higher than normal streamflow events up to 2 months ahead in an operational capacity, and up to 7 months in practice (JRC, 2018a; Arnal et al., 2018). It runs on a 5 km × 5 km grid and uses the LIS-FLOOD hydrological model (Van der Knijff et al., 2010; Alfieri et al., 2014). Seasonal ensemble* meteorological forecasts from the ECMWF's "System 4" operational meteorological forecasting system (Molteni et al., 2011) are used as input to LISFLOOD, from which seasonal ensemble hydrological forecasts are generated on the first day of each month (see Arnal et al., 2018, for details).

For the activity, SHF were produced from 1 November 2013 out to 4 months to focus on the period of extreme stormy weather and flooding experienced. As EFAS-Seasonal is designed to run at the scale of large river basins (i.e. the whole Thames basin), GIS shapefiles were used to extract forecast information for the 10 West Thames catchments using Python v3.5. This provided more locally tailored forecasts compared with Hydrological Outlook UK (Stage 1).

To ascertain whether participants had a preference for how SHF information is presented, the Stage 2 forecasts were presented as both hydrographs and choropleth* maps (Fig. 6).

Ensemble hydrographs for streamflow ($m^3 s^{-1}$) and groundwater levels (mm) indicated the predicted trajectory of the hydrological conditions for the next 4 months in each of the 10 catchments (n.b. the greater the spread, the more uncertain the forecast) (Fig. 6a). Units and dates were removed; however, exceedance thresholds*, based on daily observed streamflow and groundwater records between 1994 and 2014 for each of the catchments, were provided for context (EA, 2017; NRFA, 2017 TS3). Q50 (median) indicated average streamflow and groundwater conditions for the catchment. Q10 (90th percentile) indicated high streamflow/high groundwater level conditions – 90 % of all recorded observations over the previous 20-year period fell below this line.

The choropleth maps showed the maximum probability that the full forecast ensemble for a catchment exceeded the Q10 (90th percentile) threshold in a given month (Fig. 6b), thus providing a snapshot of the probability of potentially extreme conditions at catchment level. The full set of EFAS-Seasonal SHF provided to participants can be found in Supplement 5.

### 3.4.3 Stage 3 – "Improved" EFAS-Seasonal

Stage 3 followed the exact same set-up and provided the same style output (Fig. 7a, b) as Stage 2 – the only difference being that the seasonal meteorological forecasts used as input to LISFLOOD were taken from a set of atmospheric re-

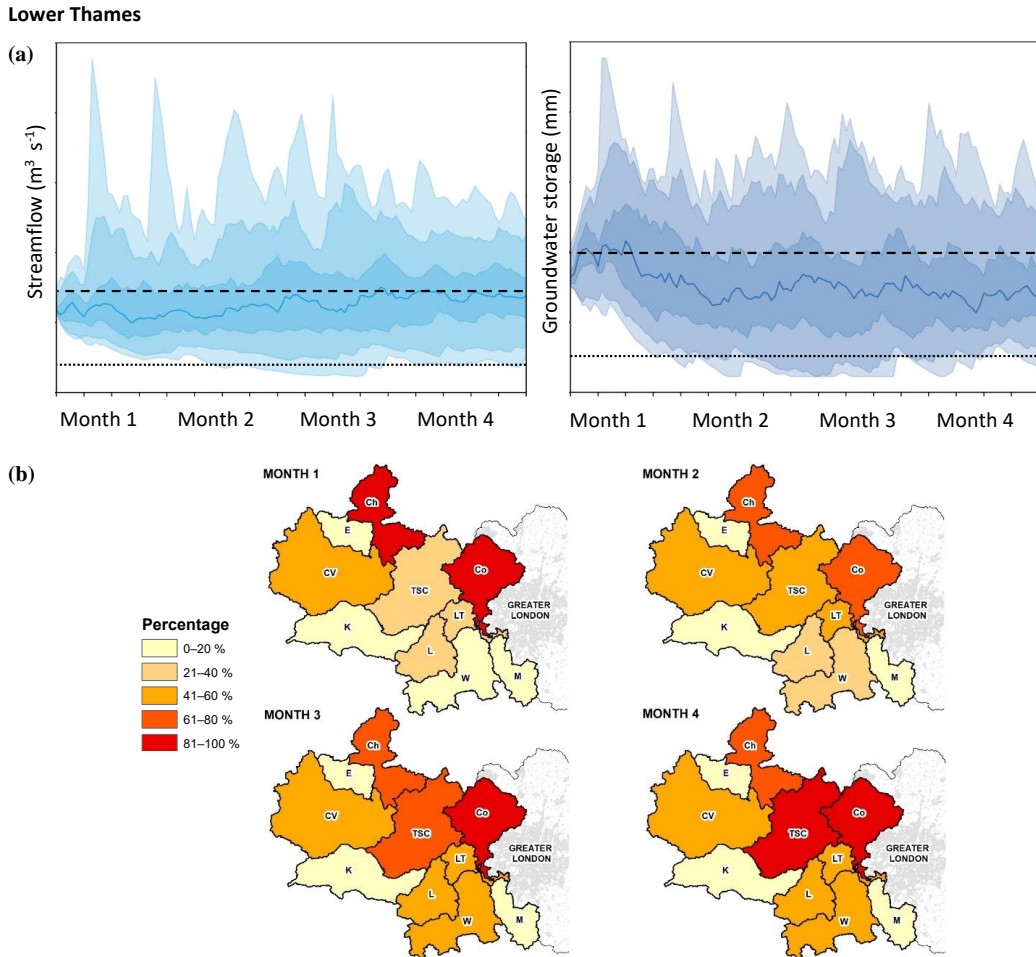

**Figure 6.** Four-month hydrological forecasts from EFAS-Seasonal (Stage 2). **(a)** Ensemble hydrographs for streamflow (light blue) and groundwater levels (dark blue) for the Lower Thames (LT) catchment. Exceedance thresholds (based on records from 1994 to 2014) are shown as Q10 (dashed line) and Q50 (dotted line). **(b)** Choropleth map shows the maximum probability that the full hydrograph ensemble for a catchment exceeds the Q10 streamflow threshold in a given month.

laxation experiments* conducted as part of a scientific study in the West Thames (see Neumann et al., 2018) rather than the operational seasonal meteorological forecasts from "System 4".

5     Atmospheric relaxation experiments were conducted by the ECMWF in late 2014 *after* the extreme weather and flooding (Rodwell et al., 2015). The aim was to recreate the atmospheric conditions that prevailed between November 2013 and February 2014, so that the ECWMF could 10 better understand how weather anomalies across the globe contributed to the flooding experienced in the West Thames (Neumann et al., 2018). The SHF at Stage 3 represented near "perfect" forecasts as they were produced *once the floods had happened and the weather conditions were known*. The 15 hydrographs are thus much sharper and more accurate than those presented to the participants at Stage 2 (Fig. 7, Supplement 6). It is important to note that this is not something that

can be achieved by operational systems currently, but does represent the theoretical upper level of forecast skill that may be available to water sector users in the future.     20

## 4 Results

### 4.1 Background context

#### 4.1.1 Catchment differences – "hydrogeology is the driving factor of risks and opportunities"

All groups recognized spatial variability between the catch- 25 ments and general consensus was that hydrogeology was the most important factor determining flood risk, drought risk, and water availability in the West Thames (Supplement 2). All groups were interested in the persistence, hydrological memory, and slower response of the groundwater- 30 driven catchments upstream (e.g. the Evenlode, Thames, and

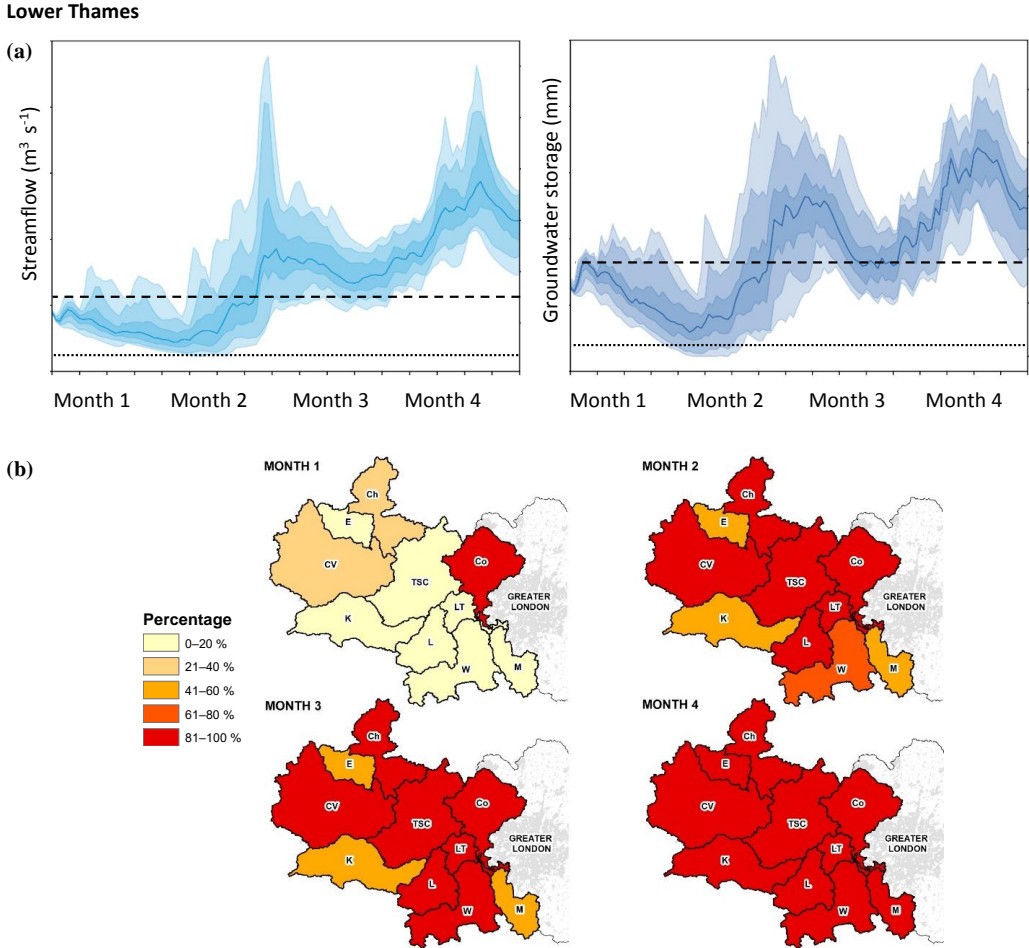

**Figure 7.** Four-month hydrological forecasts from the "Improved" EFAS-Seasonal (Stage 3). **(a)** Ensemble hydrographs for streamflow (light blue) and groundwater levels (dark blue) for the Lower Thames (LT) catchment. Exceedance thresholds (based on records from 1994 to 2014) are shown as Q10 (dashed line) and Q50 (dotted line). **(b)** Choropleth map shows the maximum probability that the full hydrograph ensemble for a catchment exceeds the Q10 streamflow threshold in a given month.

South Chilterns and Kennet) as these provided the greatest opportunity for water supply but also increased risk of local groundwater flooding and widespread fluvial flooding further downstream. Forecasters also highlighted the risks posed by impermeable catchments (e.g. the Cherwell and Lower Thames) that have a flashier response to rainfall. Water resource managers stated that upstream reservoirs were at increased risk of pollution (from agriculture), whilst dry weather (drought) was a greater issue towards London.

### 4.1.2 Current hydrological situation – "normal"

Hydrological Summary placed the "current" hydrological conditions for river flows, groundwater levels, and reservoir stocks within the "normal" range (Supplement 3). Maps indicated that rainfall was below average over the past season but above average the previous month. All groups were happy with the current hydrological situation (no risks currently), although water resource managers stated that rainfall deficiency in the background should be kept in mind due to future drought potential.

### 4.2 Participant responses from Stages 1 to 3

The findings from each stage of the activity are presented below. At no point did participants ignore the SHF information (no black stickers were placed on the maps), which matched previous discussions about organizations' current use of SHF (Sect. 2.3.2). Colour-coded decisions made by all participants (calculated by counting the stickers on the A1 catchment maps) are represented as pie charts. An accompanying bar chart details the breakdown of choices made by each participant and their specific role in the water sector (Fig. 8a–c). Quotes and information in the text are taken from discussions

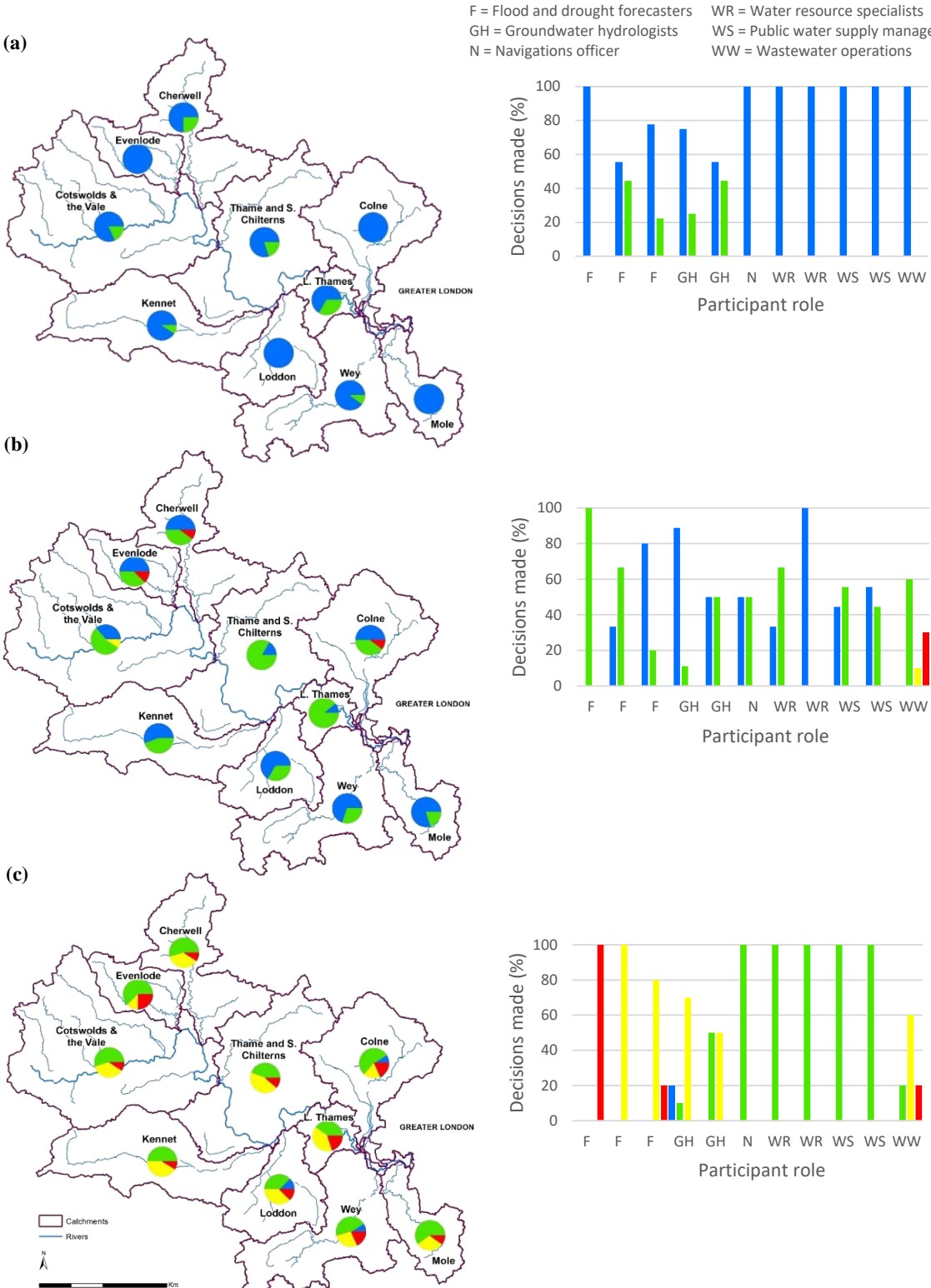

**Figure 8.** Summary of decisions and actions taken by different water sector personas based on **(a)** Hydrological Outlook UK; **(b)** EFAS-Seasonal; and **(c)** "Improved" EFAS-Seasonal. Blue – no notable risk; green – discuss internally; yellow – discuss externally and seek advice; red – implement action. Refer to Table 2 for full colour code descriptors.

recorded on the day and empathy maps – these are presented for the three groups (forecasters, groundwater hydrologists, and water resource managers).

### 4.2.1 Stage 1 – Hydrological Outlook UK

General consensus was for normal or above-normal conditions over the next 3 months; however, the information was "too vague to be actionable". Forecasters and groundwater hydrologists were more likely to discuss the situation with colleagues and keep an eye on the situation (green/blue), although there was some disagreement about the level of risk. Those involved in water resources, water supply, navigation, and wastewater operations (water resource managers) identified no risks requiring action (blue) (Fig. 8a).

Key statements TS4:

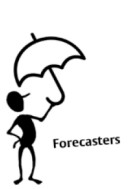

"**Analogy with the summer 2007 floods**\* suggests that **there's a risk that might be worth communicating internally**. Political influences e.g. known flooding hotspots might also be singled out for further engagement. However, there's not much evidence to divert from a normal pattern of preparedness."

\*The UK suffered extensive flooding during June and July 2007 (the West Thames was flooded in late July). Thirteen people died and damages exceeded 6.5 billion GBP TS5 nationwide (Chatterton et al., 2010).

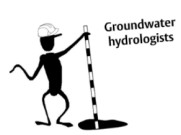

"**No major issues currently** but there is a **signal for rising groundwater levels**, potentially leading to flood risk – discuss with colleagues and keep an eye on borehole observations and new forecasts."

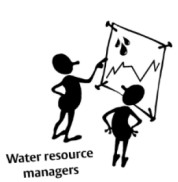

"Conditions are **favourable from a water resources perspective** – possibly heading more towards flood than drought conditions but currently **no notable risk and no concerns**. Discussions may arise during regular business briefings, but unlikely to be pursued unless changes are observed."

### 4.2.2 Stage 2 – EFAS-Seasonal

General consensus was for above-average streamflow and groundwater levels. Although the SHF provided more detail compared with Hydrological Outlook UK (Stage 1), clarity remained an issue. There was a general shift towards more internal communication (green), although actions were taken by the wastewater operations manager in the water resource managers' group (yellow/red) (Fig. 8b).

Key statements:

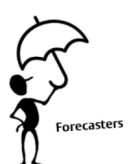

"**Repeated rainfall events can lead to accumulated flood risk** in the Lower Thames and Thame and South Chilterns. Streamflow appears to convey more risk than groundwater levels. Would discuss in general terms with colleagues and internal decision-makers to avoid an over-reaction at senior level."

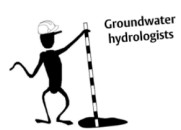

"A **moderate risk of groundwater flooding** (especially if the time period is for autumn – winter) but river flows do not appear to contribute much to groundwater risk at this stage and the forecasts are uncertain. Our **attention is focused on the chalk catchments and Thames gravels**; no direct actions are taken at the moment but we'd keep an eye on the situation and discuss at monthly meetings."

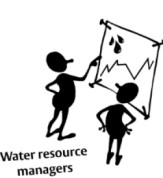

"**No significant concerns** from a water resources or navigation perspective however, there is **potential for localised flood risk which may impact on water supply and turbidity**. Not all catchments are affected so focus attention on Cotswolds and the Vale, Cherwell, Thame and South Chilterns and Colne where maps indicate high probability of Q10 exceedance. Discuss at internal briefings."

### 4.2.3 Stage 3 – "Improved" EFAS-Seasonal

General consensus was for confident forecasts that showed a high risk of streamflow and groundwater flooding in approximately 6 weeks' time. At this stage, forecasters and groundwater hydrologists were looking to verify the reliability and quality of the forecasts. Internal discussion and wider communication (green/yellow) were actively explored, although forecasters and groundwater hydrologists were still more likely to act on the information compared with water resource managers (Fig. 8c).

Key statements:

"Compared with our previous experiences of SHF these are very **sharp with a strong signal** and we would actively seek expert guidance as to the quality of the forecasts. If credible, our concern is that the signal is likely to **represent a nationwide flood risk** (not just the West Thames). **Low-consequence actions that deliver a measured message** should be implemented – e.g., identifying and locating resources and stocks, movement of temporary flood defences to high risk areas, completing projects, careful media release, strategic planning and staff briefing."

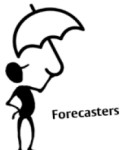

"There's **high probability of substantially exceeding the Q10 threshold**. Catchment characteristics are important to identify areas most at risk of groundwater flooding (chalk and gravels). **Drawing on previous experiences** we'd discuss the situation, obtain regular updates from partner organisations, use localised groundwater models to verify forecasts and consider communication via press release."

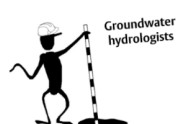

"These are **confident forecasts that give a good overview of magnitude and sequencing of possible flood events and subsequent knock-on effects to water quality**. Expect issues in 2–4 months so any actions taken would depend on how regularly forecasts are updated. We'd keep an eye on groundwater levels, hold internal briefings and discuss with groundwater team members to ensure they are kept informed and prepared. For navigation and wastewater operations where impacts can directly affect the public, we'd consider some open discussion with customers who will want to know how long an event might last."

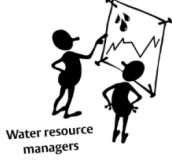
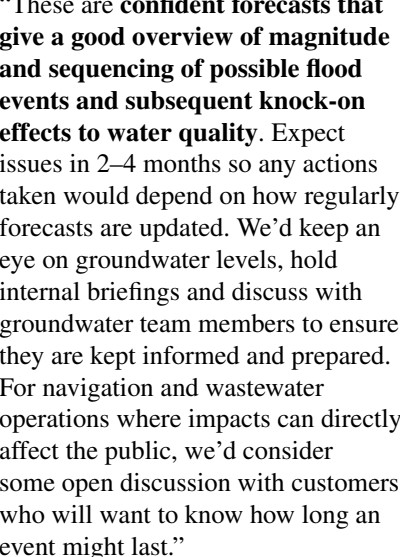

## 5   Discussion

Our decision-making activity was designed to help understand how different water sector users engage with and act on SHF at a local level. The SHF for the three activity stages rep-

resented an extreme flood event between November 2013 and February 2014. There was clear evidence that more confident (sharper) and locally tailored forecasts led to increased levels of decision and action, although water sector users did not respond uniformly. Forecasters and groundwater hydrologists were most likely to inform other organizations, request further information about the situation, and implement action, while water resource managers more consistently adopted a "watch and wait" approach. In this section, the results are discussed in more detail and the findings are placed into the wider context of policy, practice, and next steps based on discussions captured during the focus group.

### 5.1   Operational SHF systems can support decision-making and uncertainty is expected

Throughout the focus group, participants expressed positively the potential for SHF to deliver better preparedness and early warning of flood and drought events, and the benefits associated with more consistent management of water resources, whilst recognizing that low skill and coarse resolution are current barriers to use (see also Soares and Dessai, 2015, 2016; Vaughan et al., 2016; Soares et al., 2018). These benefits and barriers were demonstrated during the activity as participants increased their level of decision-making in response to the more confident and locally tailored forecasts presented: Stage 1 Hydrological Outlook UK > Stage 2 EFAS-Seasonal > Stage 3 "Improved" EFAS-Seasonal.

Hydrological Outlook UK is the first operational SHF system for the UK and was the product that participants were most familiar with, likely due to its partnership set-up (Prudhomme et al., 2017). All groups indicated that the regional focus of the maps, i.e. the whole Thames basin, and lack of resolution and certainty as to the trajectory of the upcoming hydrological conditions, limited their ability to make informed decisions. No participants however ignored or dismissed the information despite there being no perceived risk. All agreed that on a day-to-day basis, Hydrological Outlook UK serves as a useful outlook tool when supplemented with additional sources of information including water situation reports (UK Gov, 2018) and other hydro-meteorological forecasts. As of 2017, exactly how the water sector uses Hydrological Outlook UK in practice had yet to be assessed (Bell et al., 2017), and here we provide a first step towards answering this question.

Stage 2 (EFAS-Seasonal) also represented an operational forecasting system designed to run at the scale of the whole Thames basin akin to Hydrological Outlook UK. The forecasts however were presented at a catchment level on a month-by-month basis to provide a more localized outlook. This finer spatio-temporal resolution allowed participants to supplement the SHF with their knowledge of local hydrogeology and other risk factors to identify those catchments where attention would likely be most needed. This led to increased levels of communication within organizations,

even though the overall hydrological outlook was very similar to that observed at Stage 1 (uncertain but with indication towards normal–high flows). The use of large-scale (regional or global) operational forecasting products that trigger worthwhile actions at the local level has been demonstrated at shorter lead times (e.g. Coughlan de Perez et al., 2016). While the development of higher-resolution seasonal meteorological forecasts and better representation of the coupled system and initial conditions are expected to lead to improvements in SHF (Lewis et al., 2015; Bell et al., 2017; Arnal et al., 2018), we pose the open question: do operational systems such as Hydrological Outlook UK *already* have the potential to support better communication and decision-making if they could be presented at a more local scale? This would require careful communication of the uncertainty, reliability, and skill of the forecast, and how to do this effectively is a topic of current interest in meteorological and hydrological forecasting (e.g. Ramos et al., 2013; Vaughan et al., 2016; Fry et al., 2017). Although communicating uncertainty was not a specific focus of our activity, one key message from the focus group was that "uncertainty is expected" with SHF and water sector users would engage with a local forecast, even if they chose not to act on it. As pointed out by Viel et al. (2016), "low skill" is not the same as "no skill", and SHF which may have minimal value from the perspective of a scientific researcher can sometimes elicit significant interest from the view of a water sector user who is familiar with the area. Importantly, it should also be noted that although no measures of forecast skill and quality were included in our activity, participants only expressed a need to verify the quality of the forecasts at Stage 3. In discussions as to why this was the case, the forecasters and groundwater hydrologists stated that holding internal briefings and increasing awareness of "at risk" catchments are suitable low-cost actions when dealing with SHF that indicate some degree of risk, even if the information is uncertain and unverified. At Stage 3, to obtain such confident SHF was well beyond current operational standards; thus, its reliability was questioned. Participants did agree however that even in the absence of information on forecast quality, a sharper, more confident forecast that indicated high potential flood risk would be more likely to provoke a response than a dispersive one, even if the maximum of the forecast ensemble indicated values of comparable magnitude in both cases.

## 5.2  Interactions with SHF are user-specific and should be tailored accordingly

The manner in which users approached and used SHF differed markedly depending on the perceived severity of the flood event; the responsibilities and risk appetite of an organization; and the local knowledge and experiences possessed by the individual (see also Kirchhoff et al., 2013; Golding et al., 2017). Forecasters and groundwater hydrologists displayed the lowest risk appetite, admitting that they

were likely to err on the side of caution to avoid negative media impacts, economic damages, and loss of trust by the public.

> "Analogy with the summer floods of 2007 . . . my previous experience makes me think that the risk is worth communicating. . . " – forecaster at Stage 1/2.

> "A much stronger and more coherent signal regarding river flows and groundwater levels, but the forecasts indicate that the potential impact isn't right now . . . we'll keep an eye on the situation" – water resource manager at Stage 3.

While a flood event is less of an immediate issue for water resource managers, secondary effects relating to closure of canals (navigation), turbidity, and sewer surcharge (wastewater operations) did invoke action where there was potential to impact on the public. Participants were notably proactive where they had had previous experience of extreme events, e.g. forecasters' analogies with the 2007 floods (Chatterton et al., 2010), or had been witness to poor management; e.g. the wastewater operations manager recognized high potential for groundwater flooding and sewer surcharge at 1 month's lead time in the Evenlode, Cherwell, and Colne (Fig. 7).

> "Based on previous operational issues, I'd advise pre-emptive actions such as the cleaning and maintenance of pumping stations for these catchments" – Wastewater operations manager at Stage 2/3.

This highlights the value of retaining institutional memory where possible (see also McEwen et al., 2012) and being aware of organizations' or individuals' pre-determined positions or perceived self-interests which may largely be founded on previous experiences (Ishikawa et al., 2011).

It is important to note that while this activity focused on a flood event, decisions made by the groups would almost certainly have differed if the SHF had indicated drought conditions. The impacts of drought have the potential to affect larger areas, for longer (Bloomfield and Marchant, 2013), notably with respect to agriculture (Li et al., 2017), reservoir management (Turner et al., 2017) and navigation (Meißner et al., 2017). The difference in response between water sector users supports the notion that tailoring SHF information to specific user groups will improve uptake and ability to inform decision-making (Jones et al., 2015; Lorenz et al., 2015; Vaughan et al., 2016; Soares et al., 2018), an area currently being explored by the IMPREX Risk Outlook (IMPREX, 2018b).

## 5.3  Communication is both a barrier and enabler to decision-making

Communication is one of the most frequently identified barriers when it comes to uptake and use of seasonal meteorological and hydrological forecasts (Soares and Dessai, 2015;

Vaughan et al., 2016; Golding et al., 2017; Soares et al., 2018). Discussions captured during the focus group and indicated on some empathy maps identified two key communication barriers in the West Thames: (1) between water sector users themselves and how they interpret and communicate SHF information and (2) a disconnect between scientists developing the forecasts and those involved in policy, practice and decision-making.

All groups said they felt better able to interpret and communicate the messages when presented with a range of complementary forms of SHF information including maps, hydrographs, and text, with maps being of particular value. This supports findings by Lorenz et al. (2015), who identified clear differences in users' comprehension of and preference for visualizations of climate information. Mapping information was also found to be important in the survey by Vaughan et al. (2016), while numerical representations were preferred over text and graphics in the study by Soares et al. (2018). Many participants said they would feel better prepared and able to discuss upcoming hydrological conditions if SHF information was visualized in a variety of ways and regular engagement was made a routine part of their job (see Sect. 5.4).

A number of participants also felt that scientific improvements and developments to SHF are not being adequately communicated to those involved in policy and practice. General consensus was that knowledge exchange events and information sharing services through projects such as IMPREX are an excellent way of addressing this disconnect. Presentations during the focus group shared findings from other projects, including the European Provision Of Regional Impacts Assessments on Seasonal and Decadal Timescales (EUPORIAS) (Met Office, 2018), the End-to-end Demonstrator for improved decision-making in the water sector in Europe (EDgE), Service for Water Indicators in Climate Change Adaptation (SWICCA) (Copernicus, 2017a, b), and Improving Predictions of Drought for User Decision Making (IMPETUS) (Prudhomme et al., 2015) – much of which was new knowledge to some participants. It was further expressed that stakeholder events yield maximum benefit for both the scientist and the user when they are co-produced with an organization that is involved in receiving, tailoring, and distributing SHF information (Rapley et al., 2014). Importantly, we do not want to be in the position whereby SHF skill has improved but the credibility and reliability of the information is questioned by decision-makers who have not been kept up to date with developments. The potential for this disconnect was demonstrated by both forecasters and groundwater hydrologists at Stage 3 ("Improved" EFAS-Seasonal) whereby decisions would only be made if the accuracy of the forecast could be verified.

> "Forecast signal is implausibly strong but, if valid, gives a clear signal for disturbed conditions"

> "Surprised at forecast and the strength of the signal… IF credible, then actions need to be taken"

"Would definitely talk to the Environment Agency and search for other monitoring data to verify the forecast" – forecasters and groundwater hydrologists at Stage 3.

In this case, the SHF at Stage 3 were hypothetical and no information on forecast quality was given; however, the forecasts provided a good representation of what scientists hope to achieve with operational seasonal forecasting systems in the future (Neumann et al., 2018). This emphasizes the need to keep water sector users informed of scientific developments (see also Bolson et al., 2013), and to build awareness and knowledge around interpreting and using forecast quality information, as it is becoming more widely adopted in seasonal forecasting (see Copernicus, 2017a; Fry et al., 2017).

## 5.4 Implications for future policy and decision-making

The EA is the public body responsible for managing flood risk in the UK. They focus on maintaining a certain level of preparedness whilst recognizing that particular conditions and types of flooding/drought are more likely at different times of year. Currently, the EA use SHF predominantly as supporting information and rely on shorter-range forecasts for action. As co-developers of this focus group, the EA recognized the following points for future consideration.

1. To upskill and help staff interpret SHF information received.

2. To identify suitable low-consequence actions that could be taken based on SHF.

3. To move beyond the current position of using SHF for information only, to making conscious decisions as part of routine incident management strategies (relies on 1 and 2).

> "Regular review and discussion of extended outlooks (5–30 days) and the 1–3 months forecasts during weekly handover between the incoming and outgoing flood duty teams would improve familiarity of long range forecast products and dealing with the uncertainty that they present. This would be an excellent way of considering the possible conditions and the potential for disruption going forward." – EA activity co-developer.

In short, more engagement with SHF and improved clarity for easier interpretation by different users will ensure that SHF have a valuable role to play in future decision-making at the local scale.

## 5.5 Learning outcomes and future considerations

Encouragingly, we identified that SHF are being used, and participants agreed that the decision-making activity was an

entertaining platform for fostering discussions which complemented their everyday work and general understanding of SHF. From the participants' perspective, learning outcomes included knowing more about the ongoing scientific developments in SHF and a better understanding of how different organizations in the West Thames water sector are using SHF. Many also stated that the activity and focus group discussions enhanced their ability to think about possible decisions and actions that may be taken in the future. As the activity developers, we found that the group discussions stimulated participants' motivations and interests more so than would have been achieved by asking participants to engage on an individual basis. We also advocate the use of empathy maps or other forms of obtaining a written record of participant thought processes in addition to their decision choices.

Our activity was designed to provide a first insight into the current state of play regarding SHF in the West Thames. Although 11 participants was a small sample size, they represented an important and well-balanced mix of water sector decision-makers in the West Thames. The only exception was the agricultural sector, which could not attend, and thus it would be interesting to capture this perspective with ongoing research (e.g. Li et al., 2017). We also recognize the possibility that those who took part had a vested interest in SHF; however, we did encourage participants to attend even where they had no background knowledge or experience of SHF. Finally, we advocate that others conducting a similar activity may wish to consider whether participant interpretation can be subconsciously influenced by the information provided. For example, flood risk maps were provided as part of the background context, but may have inadvertently led participants to consider the upcoming forecasts with respect to high-flow events. Likewise, there is potential that the 3-month SHF (Stage 1) may have been interpreted differently to the 4-month forecasts (Stage 2 and Stage 3) and we do not know the degree to which individuals may have been swayed to place a particular colour on the map based on the conversations they had with their group members (and how big an influence such conversations play in real life). Discussions with the participants at the end of the activity with respect to these points would have been helpful.

## 6 Conclusions

Key findings were that engagement is user-specific and SHF have the potential to be more useful if they could be presented at a scale which matches that employed in decision-making. The ability to interpret messages is aided by complementary forms of SHF visualization that provide a wider overview of the upcoming hydrological outlook, with maps being of particular value. However, improved communication between scientists, providers, and users is required to ensure that users are kept up to date with developments. We conclude that the current level of understanding in the West Thames provides an excellent basis upon which to incorporate future developments of operational forecasts and for facilitating communication and decision-making between water sector partners.

**Data availability.** All data/graphs/information that were used by participants for the focus group activity are included in the Supplement. Individual participant results are not publicly available in order to protect anonymity. If readers require further information, this may be provided by contacting the corresponding author.

## Appendix A: Glossary

| | |
|---|---|
| Aquifer | underground layer of water-bearing permeable rock which can occur at various depths. |
| Atmospheric relaxation experiments | are used by meteorologists once an extreme weather event has happened. Put simply, when a seasonal forecast predicts the wrong weather, scientists "force" the conditions in the atmosphere so that they can try to recreate the extreme weather conditions and better understand what happened. |
| Baseflow | the portion of the river flow (streamflow) that is sustained between rainfall events and is fed into streams and rivers by delayed shallow subsurface flow. Not to be confused with "groundwater" which is water which has entered an aquifer, or "groundwater flow" where water enters a river having been in an aquifer. |
| Choropleth map | uses differences in shading, patterning or colouring in proportion to the value of a given variable in areas of interest. |
| Exceedance threshold | a user-defined threshold (e.g. 90 %) that is based on river flow or groundwater level observations (measurements) from the previous 20 years. E.g. if an exceedance threshold is set to the 90th percentile, this means that 90 % of all recorded observations over the past 20 years fell below this level. |
| Flashy | rivers and catchments that respond quickly to rainfall events. |
| Forecast ensemble | instead of running a single forecast (known as a deterministic forecast that has one outcome), computer models can run a forecast several times using slightly different starting conditions (to account for uncertainties in the forecasting process). The complete set of forecasts is referred to as the ensemble, and the individual forecasts are known as ensemble members. Each ensemble member represents a different possible scenario, and each scenario is equally likely to happen. |
| Forecast quality | the SHF is compared to, or verified against, a corresponding observation of what actually happened, or a good estimate of the true outcome. SHF quality describes the degree to which the forecast corresponds to what actually happened (see also "forecast skill"). |
| Forecast sharpness | describes the spread or variability among the different ensemble members of a forecast (the different forecast values). The more concentrated (close together) the ensemble members are, the sharper the forecast is, and vice versa. Importantly, a forecast can be sharp even if it is wrong i.e. far from what actually happened. (See also "forecast ensemble".) |
| Forecast skill | the SHF quality can be compared to the quality of a benchmark or reference, usually another forecast. The relative quality of the SHF over this reference forecast is the SHF skill (see also "forecast quality"). |
| Forecast uncertainty | the skill and accuracy of SHF tends to decrease with increasing lead time due to factors such as variations in weather conditions, how the hydrological model has been set-up to represent complex processes, and how well the hydrological model has captured the real-world hydrologic conditions at the time the forecast is started (e.g. how wet is the soil or how much water is currently in the river?). There is an element of uncertainty in all forecasts that can amplify with time. Ensemble forecasting is one way of representing forecast uncertainty. (See also "forecast ensemble".) |
| Hydrogeology | the area of geology that deals with the distribution and movement of below-ground water in the soil, rocks and aquifers. |
| Hydrograph | a graph showing how river and groundwater levels are expected to change over time at a specific location. Ensemble hydrographs show the full spread of the forecast ensemble. |
| Lead time | the length of time between when the SHF is started (initiated) and the occurrence of the phenomena (e.g. flood) being predicted. Can also be used to represent the point at which the SHF is started and the beginning of the forecast validity period (e.g. from 3 weeks). |
| Lithology | the general physical characteristics of rocks. |
| River basin | the largest and total area of land drained by a major river (in this case the River Thames) and all its tributaries. (See also "river catchment".) |
| River catchment | the area of land drained by a river. "Catchment" and "basin" are sometimes used interchangeably. Here catchments represent the drainage areas of the River Thames main tributaries, of which there are 10 in the West Thames. |

| Seasonal hydrological forecasts (SHF) | provide information about the hydrological conditions e.g. streamflow (river flows), ground-water levels and soil moisture levels, that might be expected over the next few months (e.g. from 3 weeks out to 7 months). |
|---|---|
| Seasonal meteorological forecasts | provide information about the weather conditions e.g. rainfall, air temperature, humidity, pressure, wind, that might be expected over the next few months (e.g. from 3 weeks out to 7 months). |
| Streamflow | the flow of water in a stream or river. Also known as river flow. |
| Surface runoff | the flow of water that occurs when water from excess rainfall, meltwater or drainage systems flows over the Earth's surface and not into the ground. |
| Tributary | a river or stream that flows into a larger stream, river or lake. Tributaries do not flow into the sea. |
| 1 : 100-year flood event | a 100-year flood is a flood event that has a 1 % chance of occurring in any given year. |
| 1 : 5-year flood event | a 1-in-5-year flood is a flood event that has a 20 % chance of occurring in any given year. |

## Information about the Supplement

– Supplement 1: Invitation flyer and programme for the focus group

– Supplement 2: West Thames catchment characteristic maps

– Supplement 3: Hydrological Summary: October 2013, June–September 2013 and November 2012–October 2013

– Supplement 4: Stage 1 Hydrological Outlook UK: November 2013–January 2014

– Supplement 5: Stage 2 EFAS-Seasonal: November 2013–February 2014

– Supplement 6: Stage 3 "Improved" EFAS-Seasonal: November 2013–February 2014

**Supplement.** The supplement related to this article is available online at: https://doi.org/10.5194/gc-1-1-2018-supplement.

**Author contributions.** JLN and LA designed the decision-making activity. JLN, LA, SH, and HLC co-organized the set-up of the focus group. All the authors took part in delivering the focus group, including as note-takers, organizers, and presenters of their scientific research. JLN wrote the manuscript with input from all the authors.

**Competing interests.** The authors declare that they have no conflict of interest.

**Disclaimer.** The information and findings in this paper are based on discussions and actions captured during the decision-making activity. They should not be taken as representing the views or practice of particular organizations or institutions.

**Acknowledgements.** This work was funded by the EU Horizon 2020 IMPREX project (http://www.imprex.eu/, last access: 21 May 2018) (641811) with additional financial support provided by the University of Reading's Endowment Fund. Support-in-kind was also provided by the NERC LANDWISE project (https://landwise-nfm.org/about/, last access: 10 July 2018) (NE/R004668/1). We would like to express our sincere thanks to all participants who shared their knowledge and experience relating to seasonal hydrological forecasting and to their organizations who enabled their participation. We would especially like to thank Stuart Hyslop and Simon Lewis at the EA for their support in the organization of the day and also Len Shaffrey (Department of Meteorology, University of Reading) for his input on the day.

Edited by: Katharine Welsh
Reviewed by: two anonymous referees

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

## Remarks from the language copy-editor

CE1     Please note that it would be better to leave this name without a fullstop, as this is how it appears on the Met Office's website.

## Remarks from the typesetter

TS1     We we like to leave this as is since this is our standard unit. If this is incorrect, we would be very grateful if you could provide further explanation regarding this issue.

TS2     Please confirm.

TS3     Please confirm.

TS4     Regarding the layout of this section, please note that text always has to be in the two-column layout. Another possibility would be to put the figures and text in a table environment (which could then span over two columns.). Please note that tables cannot be inserted in between text. The tables would have to be placed at the beginning of a page and could therefore not be placed under a specific paragraph. Please let me know how you would like to proceed with this issue.

TS5     Please note that, to ensure a transparent review process, this correction will have to be confirmed by the editor. Please provide a short statement explaining this change which will be forwarded by us to the editor. Thank you very much in advance.

TS6     Please note that you had inserted the reference to the discussion version of this paper. The final revised version has been published in the meantime and the reference has been updated during the typesetting.