# Peer review of "Can seasonal hydrological forecasts inform local decisions and actions? A decision-making activity"

_Geoscience Communication, 2018_

## Referee Comment (RC1) · Anonymous Referee #1 · 27 Aug 2018

The paper is clearly written and well-structured. I think however that it needs some major clarifications.

Overall, my main issue in the paper is that while the results (page 18, lines 4-5 summarize the results very well) are interesting, they are not novel (or surprising). It seems to me that decisions tend to reflect the level of information provided for the decision-making process: too vague information prompted less actions than sharper (even if not of better skill) information. If the focus is on providing evidence of how decisions depend on the way information is being communicated/visualized, then this should be stated already in the introduction of the paper, together with the aims of the paper. I

think this would add value to the findings. The stated objective of the paper ("to develop a clearer understanding. . .", line 5, page 3, in order to learn "weather SHF can be used to support decision-making at local level", line 23, page 5) seems to be too vague with respect to the actual findings highlighted in the results part of the paper.

Also, I have some doubts on the validity of the use of the "empathy map". My detailed comments are provided in the list of comments below.

Introduction (pages 2-3):

- I was sometimes surprised by the references that were inserted at some places. I had sometimes the impression that the references mentioned in brackets were not really related to the paper or the subject just mentioned. For instance, when the authors mention EFAS and cite Bartholmes et al., 2009 and Smith et al., 2016, they are citing papers related to medium-range flood forecasts and not seasonal hydrological forecasts (SHF), which is the main topic of the paper and of the paragraph ("SHF systems covering a range of spatial scales"; lines 20-21). Another example is in line 27, where the authors mention papers "exploring how users engage with and apply SHF to inform decisions", but cite Demeritt et al., 2011 and Ramos et al., 2013, which also deal with medium-range forecasts and not seasonal range forecasting. The same is observed in line 31, where studies not related to seasonal forecasts are cited as "others investigating the application of seasonal meteorological forecasts", or again on line 25, where skill of SHF in Europe is referenced by Doblas-Reyes et al., 2013, which is a paper dealing with meteorological forecasts and predictions and not hydrological forecasts. Overall, my impression is that the authors are not citing the papers correctly, or at least it is not clear or straightforward for the reader. I suggest them to review their citations.

- I also suggest the authors to be more explicit on the content they want to highlight from the literature they cite. For instance, lines 19-20, when they say that "recent research has demonstrated improvements", it is important to be more explicit about what improvements are these: gain in skill, accuracy, less errors? How much percent

of improvement? Also, in line 30, I think it is not clear why "uncertainty" is a "key barrier to use". For long now, operational hydrologists and water managers have used probabilistic or scenario-based predictions for seasonal forecasts (see, for instance, the use of ESP, Ensemble Streamflow Predictions, which dates back to the 1980s). Firstly, I think the barrier is not on using seasonal predictions, but GCM-based forecasts, which is quite different. Secondly, I think the barrier is not the uncertainty, but the fact that GCM-based forecasts do not always show better skill than ESP-based forecasts. At some places of the world, they do not do better than the "easier" (if we can say so) use of climatology of precipitation and temperature plus a hydrological model as in the ESP technique. I suggest the authors to clarify what they call a barrier: if no seasonal information is used at all by stakeholders in the West Thames, then the reasons may be other than just uncertainty or decreasing skill (for instance, it could be institutional barriers or lack of perception of its usefulness).

- Also in the Introduction, I miss a more explicit justification for the study. The motivation is quickly referred to in lines 21-23, but it mixes flood warning, drought risk, and water resources management. It is not clear to me why seasonal forecasting is the focus here. I do not think it is straightforward to think of the usefulness of seasonal forecasts for capturing a "flood event with 3 months lead time" (line 17). How many (and which) users rely on a 3-month information to forecast a flood event and take action? Unless we refer to very large catchments, with several months of response time, floods are rather characterized as more rapid phenomena, which are already hard to forecast with accuracy in short to medium-range forecasting systems. I think that the use of seasonal forecasts to forecast a flood event should be better justified in the paper (we note also that the authors even mentioned "flashy response to storm" in the area of the study in line 8). Is it an "expectation" of the users in the West Thames? What are the actual users' needs and expectations in the case presented in the paper?

- Finally, also in the Introduction, I think line 7 "expert flood science communication" should be clarified: what do you mean by this terminology? Also, the "in-the-moment

activity" should be clearly defined, especially as it is used in the title of the paper. I can understand "real-time", but "in-the-moment" is not fully clear to me.

Page 5: - Lines 20-24: I suggest rewriting this part as it is a bit confusing: why is it important to learn what the authors assumed about previous knowledge of participants? Finally, isn't it only important what they finally got at the end in the focus group? - Line 24: what are the differences between raw forecasts and the Hydrological Outlook UK? - Line 29: I would be curious to know how/why the users think SHF has potential to forecast floods. I can understand that it may indicate if the season will be drier or wetter than normal, but this is not necessarily an indication that a flood event will happen and action should be taken now, for instance. This is, at least, not a common way of using SHF information. - Line 31: what is a "coarse" spatio-temporal resolution to them?

Page 6: - Line 29: I think it is a bit disturbing not to show units. I guess the hydrologists have a feeling that increasing from 1 m3/s to 2 m3/s, for instance, may not be as impacting as increasing from 1 m3/s to 30 m3/s. Graphs without units may be tricky and convey "distorted" information: we have the impression of seeing an important increase in flows when, in fact, it does not represent much, or vice-versa. Have you had reactions in that sense from the participants? How could them place their forecasts with respect to climatology, current hydrological situation, background info, etc. without the "units"? Were these pieces of information also given without units? (see also page 9, lines 4-5) - Line 30: if no information on forecast quality was provided, how could the users know that Stage 3 sharper forecast were more skilful (how could they evaluate if the forecasts from all stages were, first of all, reliable?) - Line 31: why the information was not given for 3 month lead time in all situations? Do you think that the fact of having two systems providing longer lead time information may have influenced the appreciation users might have of the quality of the systems?

Pages 8-9: - A traditional empathy map is usually used to describe what a type of user think & feel, hear, see, say & do, and also gather their typical "fears" and "gains" in order to create a shared understanding of user's needs. I did not understand well how
it was employed here: what were the objectives and how did you build it based on the concepts of the traditional approach? How did you move from the "individual" to the "collective"? From Fig. 4, I have the impression that there are two aspects being "surveyed" in the map: the SHF and the decision. How have you separated these in the analysis? Finally, was the concept behind empathy map really needed? I have the impression that the map used here differs significantly (in its target) from the traditional one, which is mostly employed to establish a common ground among team members and to understand and prioritize user's needs. If I understood correctly, basically, a traditional empathy map is built by a group of individuals to reflect a certain type of user, while here it is built by individuals to reflect their own individual preferences, which are them put together to represent a group (although it is not said how this analysis is done in the paper). I also could not understand how "discussions were captured on the empathy maps" (as stated in line 4, page 20). Could you comment on these issues? - While the situation of high flow or low flow was not informed in the beginning (as mentioned by the authors on page 6, line 28), info on flood risk only was provided (page 9, line 13). Isn't there a chance that this has influenced the participants to consider the forecasts as that of an upcoming flood event? Also, maps were provided showing exceedances of given Q thresholds, which is more common in floods than drought visualization maps. Finally, was it really necessary to hide that he forecasts were for a potential flood situation? Wouldn't they guess it anyway? - Page 9, line 17: do you mean "past (or previous) month"? - Page 18, line 13: I think "low skill" and "more dispersive" forecasts are interchangeably used here. The fact that stage 3 forecasts were sharper does not mean they were more skilful, although the perception of the participants seems to be that they were. I think this needs a more clear explanation. Users may think that the sharper, the better, which is not always the case in SHF. From my experience, water managers tend to understand this aspect much better than flood forecasters.

Discussion: - Some parts seem to present results and not a discussion. For instance, page 20 presents, for the first time, the results of the empathy maps analysis. I think

this should go to the section on results. By mixing these sections (results and discussion) it is unclear what pertains to the stakeholders' points of view and what comes from the authors' views/perceptions. For instance, when mentioning a series of projects, IMPREX, EUPORIAS, Edge, SWICCA, etc., it is not clear if these projects were discussed with the stakeholders and participants of the activity reported in the paper (and how it influenced, or not, the discussions and responses to the activity). - Page 21, lines 15-16: it is not indicated anywhere in the methodology of the activity where knowledge on "ongoing scientific developments in SHF" was conveyed to participants. This "conclusion" seems to be disconnected with the methodology presented in the paper. - Page 21, lines 19-21: I am a bit annoyed here because of the original concept behind "empathy maps", as mentioned earlier in my remarks. The definition of "empathy", itself, is related to the action of understanding, being aware of, being sensitive to, etc. the "other" (not oneself), while here it is used by the participants to express their own feelings. They use it as a mirror and not as a way to "put yourself in someone else's shoes". My question is: is it really the appropriate tool to address the issues you want to address? Why did the authors choose this specific tool? I believe the authors got the answers to their questionings, but I have the feeling that this was done with a tool that was fully adapted by them to that purpose, and not as a result of applying the well-known technique of the "empathy map". I think this needs to be clarified.

---

## Referee Comment (RC2) · Anonymous Referee #2 · 3 Sep 2018

*General comments*

I would like to thank the authors for this innovative contribution that addresses a still little-explored topic in the (seasonal) hydrological forecasting research community. The proposed article explores forecast uncertainty, forecast communication and decision-making with an original experiment involving and giving voices to decision-makers from the West Thames. As highlighted by the authors, this is a research area with limited contributions, which, in my opinion, fits nicely in Geosciences Communication. The paper is didactic and well documented, and I strongly recommend it for publication, though I do have some minor questions which I list below.

[Figure]

\*Specific comments\*

Section 2.2.1: (a) Here, I think some more information on the process of finding participants would be useful for the reader. From this section, it seems that all eleven invited participants agreed to be part of the focus group: "11 West Thames stakeholders [...] were invited to take part in the focus group [...]" (L.12). If so, I assume there were previous collaborations, and did these have any role in the willingness to participate? And if not, how many stakeholders were invited, how many declined and, if any, for which reasons? (b) Additionally, it would be interesting that the authors mentioned how many forecasters, public water suppliers, waste water modellers and operators, etc. are active in the area. For example, how representative are the 3 forecasters that took part in the group? What do 11 stakeholders represent at the scale of the region? This is for the sake of giving a wider picture to the reader on the stakeholders being active in the area, and on the impact/outreach this experiment has had. (c) How many different organizations were represented through these 11 participants?

Section 3.2.2: How much do you think the group opinion influenced the colour chosen by individual participants? Could there be biases here?

Figure 3: It is mentioned in the text that there are about 110 dots on each map (L.9), but in Figure 3, we observe around 9 or 10 dots per catchments. What is the reason for this? Were some stakeholders only working on their usual catchments of interest?

Section 3.3.1: It was not clear to me how familiar the participants were to this information and what they got out from it if they are already familiar with the region.

Section 4.2: "At no point did participants ignore the SHF information" (L.18) Isn't this result due to the context? From the moment the participants know they are in a seasonal forecasting experiment, they are willing to use the provided information.

Section 5.3 (L.22-29): How did the authors deal with forecast quality in this experiment? (a) From this paragraph, it seems that no quality information was provided, and

indeed, no quality information appears in the Stage 1, Stage 2 and Stage 3 sections of the Supplement. Was it a choice to exclude this information, or is it not available to users in the Hydrological Outlook UK and the EFAS-Seasonal? (b) In the absence of quality information, did the users assume that forecast quality was the current/latest one they are aware of? (c) Several platforms now propose quality information along with the forecasts, and assuming that this information is provided in a clear manner, users do not have to hypothesize about the quality of the forecasts they use. In this specific paragraph, authors suggest "to keep water sector users informed of scientific developments". In my opinion, it is also crucial to provide quality information in an intelligible way along with the forecasts, as well as build the required user knowledge to understand this information.

*Technical corrections*

Figure 8: A reminder of the colour codes would be useful for the reader to have.

---

## Editor Comment (EC1) · K. Welsh (Editor) · 5 Sep 2018

I would like to thank both reviewers for their prompt reviews of this manuscript. The discussion remains open until 19th September for further comments. I look forward to responses and an updated version of the manuscript from the authors.

---

## Author Comment (AC1) · 12 Sep 2018

Dear Katharine and reviewers,

Thank you very much for all your comments. We are now working through these and will post a full response with our updated manuscript very soon.

With thanks

Jess and co-authors
* * *

---

## Author Comment (AC2) · 21 Sep 2018

We thank referee 1 for taking the time to review our manuscript and for providing very thorough and detailed comments. We've improved our manuscript accordingly and our responses to all the reviewer's comments are included below. We have used the following sequence: (1) comments from Referee, (2) author's response, (3) author's changes in manuscript

(1) The paper is clearly written and well-structured. I think however that it needs some major clarifications.

[Figure]

Overall, my main issue in the paper is that while the results (page 18, lines 4-5 summarize the results very well) are interesting, they are not novel (or surprising). It seems to me that decisions tend to reflect the level of information provided for the decision-making process: too vague information prompted less actions than sharper (even if not of better skill) information.

(2) Thank you for these comments. You are correct that the decisions did tend to reflect the level of information provided and that this would be broadly expected with more confident forecasts. However, before the activity was conducted, we did not know how (or why) different user groups would respond, as knowledge about how organisations are using SHF in the West Thames was relatively unknown. By asking them to play in their day-job roles, there was the possibility that participants might ignore (or not act on) any of the information provided. Rather, the results highlighted that Hydrological Outlook UK (Stage 1) is the operational SHF that users are currently engaging with and this is something that has not really been looked at in the UK to date. The wish to verify the reliability of the Improved-EFAS (Stage 3) SHF also highlighted that there is a preconception about the level of confidence currently associated with SHF. Overall the findings indicated that users respond differently and that better decision-making could potentially be made by presenting locally-tailored SHF information that is visualised in different ways. This raises important questions about whether it is possible to make operational products such as Hydrological Outlook UK 'less vague' and more useable by taking account of these points.

(1) If the focus is on providing evidence of how decisions depend on the way information is being communicated/visualized, then this should be stated already in the introduction of the paper, together with the aims of the paper. I think this would add value to the findings.

(2) How to visualise information was not a key focus of what we were looking at originally, however, it did become clear throughout the activity that participants responded favourably to having information presented in a variety of ways (especially maps).

Interactive
comment

(3) We have updated the abstract (Page 1, line 26 – 29) to incorporate this finding.

(1) The stated objective of the paper ("to develop a clearer understanding: : :", line 5, page 3, in order to learn "weather SHF can be used to support decision-making at local level", line 23, page 5) seems to be too vague with respect to the actual findings highlighted in the results part of the paper.

(2) Thank you. We have included a much clearer justification about why we conducted the focus group and activity taking account of what we know about water sector needs and expectations (supported by the Environment Agency and what they'd like to know), plus also why the West Thames is a valuable case study site that would benefit from regular engagement with SHF.

(3) Please see the new paragraph on page 3, lines 5 – 15.

(1) Also, I have some doubts on the validity of the use of the "empathy map".

(2) We have commented on this below.

(1) My detailed comments are provided in the list of comments below.

Introduction (pages 2-3): I was sometimes surprised by the references that were inserted at some places. I had sometimes the impression that the references mentioned in brackets were not really related to the paper or the subject just mentioned. For instance, when the authors mention EFAS and cite Bartholmes et al., 2009 and Smith et al., 2016, they are citing papers related to medium-range flood forecasts and not seasonal hydrological forecasts (SHF), which is the main topic of the paper and of the paragraph ("SHF systems covering a range of spatial scales"; lines 20-21). Another example is in line 27, where the authors mention papers "exploring how users engage with and apply SHF to inform decisions", but cite Demeritt et al., 2011 and Ramos et al., 2013, which also deal with medium-range forecasts and not seasonal range forecasting. The same is observed in line 31, where studies not related to seasonal forecasts are cited as "others investigating the application of seasonal meteorological forecasts",

or again on line 25, where skill of SHF in Europe is referenced by Doblas-Reyes et al., 2013, which is a paper dealing with meteorological forecasts and predictions and not hydrological forecasts. Overall, my impression is that the authors are not citing the papers correctly, or at least it is not clear or straightforward for the reader. I suggest them to review their citations.

(2) Thank you for highlighting that this might not be clear to the reader. Our original intention was to direct readers who are interested in finding out more to relevant sources (e.g. about the EFAS setup or using hydrological forecasting for decision-making in general). However we appreciate that this may be confusing as not all sources related to SHF.

(3) We have rephrased our text and amended our citations so that it is clear exactly what each citation will relate to.

(1) I also suggest the authors to be more explicit on the content they want to highlight from the literature they cite. For instance, lines 19-20, when they say that "recent research has demonstrated improvements", it is important to be more explicit about what improvements are these: gain in skill, accuracy, less errors? How much percent of improvement?

(2/3) Thank you – the papers cited cover quite a few improvements but we have now mentioned that which we feel is most relevant here, specifically the increased accuracy out to 4 months of lead time for high flow events during the winter months in Europe.

(1) Also, in line 30, I think it is not clear why "uncertainty" is a "key barrier to use". For long now, operational hydrologists and water managers have used probabilistic or scenario-based predictions for seasonal forecasts (see, for instance, the use of ESP, Ensemble Streamflow Predictions, which dates back to the 1980s). Firstly, I think the barrier is not on using seasonal predictions, but GCM-based forecasts, which is quite different. Secondly, I think the barrier is not the uncertainty, but the fact that GCM-based forecasts do not always show better skill than ESP-based forecasts. At some

places of the world, they do not do better than the "easier" (if we can say so) use of climatology of precipitation and temperature plus a hydrological model as in the ESP technique.

(2) Thank you for this. By 'uncertainty' we are meaning the fact that the skill of SHF (whether from ESP or GCM methods (see Arnal et al., 2018)) tails-off at longer lead-times (beyond 1 or 2 months) which can affect the perceived usability and reliability of the information by users. This is clarified on Page 2, line 30.

While we agree that there can be differences in the skill and accuracy of GCM-based forecasts and ESP-based forecasts (depending on location, season, lead-time), we don't know how much users are aware of this technical information and so we would like to retain our current clarification that 'uncertainty' refers to forecast skill and sharpness decreasing with increasing lead time. Providing this level of detail is in line with other studies e.g. Soares and Dessai, 2015 and we believe suitable for Geoscience Communication where readers may not have hydrological backgrounds.

Refs: Arnal et al., 2018 - Skilful seasonal forecasts of streamflow over Europe. Soares & Dessai, 2015 - Barriers and enablers to the use of seasonal climate forecasts amongst organisations in Europe.

(1) I suggest the authors to clarify what they call a barrier: if no seasonal information is used at all by stakeholders in the West Thames, then the reasons may be other than just uncertainty or decreasing skill (for instance, it could be institutional barriers or lack of perception of its usefulness).

(2) Yes. We please refer the reviewer to the top of Page 3 where we present other factors that may be barriers to use including how SHF is visualised, the levels of communication in the water sector and the level of knowledge and training required to interpret and apply SHF information effectively. For the West Thames users, we discuss these factors in detail in the Discussion section.

(1) Also in the Introduction, I miss a more explicit justification for the study. The motivation is quickly referred to in lines 21-23, but it mixes flood warning, drought risk, and water resources management. It is not clear to me why seasonal forecasting is the focus here. I do not think it is straightforward to think of the usefulness of seasonal forecasts for capturing a "flood event with 3 months lead time" (line 17). How many (and which) users rely on a 3-month information to forecast a flood event and take action? Unless we refer to very large catchments, with several months of response time, floods are rather characterized as more rapid phenomena, which are already hard to forecast with accuracy in short to medium-range forecasting systems. I think that the use of seasonal forecasts to forecast a flood event should be better justified in the paper (we note also that the authors even mentioned "flashy response to storm" in the area of the study in line 8). Is it an "expectation" of the users in the West Thames? What are the actual users' needs and expectations in the case presented in the paper?

(2/3) Thank you, these are interesting points that you have raised and we have now included a much more explicit justification for the study to help answer these (page 3, lines 5 – 15). Below are some of the key points we have taken to address your comments:

We discuss the West Thames in terms of its hydrogeological setup i.e. it is a largely groundwater-driven system that has a slow hydrological response which has relevance for flooding, drought and water resource management at seasonal timescales. We agree that while some flood events in the West Thames (especially those caused by sudden intense rainfall) are rapid phenomena, there is evidence that prolonged rainfall events can lead to fluvial and groundwater flooding which can be detected using SHF. We have included references to support this justification for both flood and drought outlook in the Thames basin (page 3, lines 8-9).

We did not know the needs or expectations of the users in the West Thames with respect to SHF (this was a key output of the activity). However, we have highlighted why using SHF is of interest to the UK water sector (Prudhomme et al., 2017) and

outlined the expectations of the Environment Agency in terms of what they hoped to learn from the activity (lines 5 and 11).

On page 3, lines 33-35 we have also included further justification explaining that the activity, although forecasting for a flood event, also encouraged discussion with respect to associated impacts on water quality, sewage treatment and water resource management.

(1) Finally, also in the Introduction, I think line 7 "expert flood science communication" should be clarified: what do you mean by this terminology?

(2) We have rephrased to say "In the context of flood science communication with experts".

(1) Also, the "in-the-moment activity" should be clearly defined, especially as it is used in the title of the paper. I can understand "real-time", but "in-the-moment" is not fully clear to me.

(2) We have removed the term "in-the-moment" and used "real-time" (the title now just reads as "a decision-making activity").

(1) Page 5: - Lines 20-24: I suggest rewriting this part as it is a bit confusing: why is it important to learn what the authors assumed about previous knowledge of participants? Finally, isn't it only important what they finally got at the end in the focus group?

(2) This sentence has been removed as we agree that it is a bit confusing.

(1) Line 24: what are the differences between raw forecasts and the Hydrological Outlook UK?

(2) The Hydrological Outlook UK is a forecast summary taken from a set of meteorological, hydrological and analogue models.

(3) We have rephrased to say:

"...participants noted that Hydrological Outlook UK (CEH, 2018) and the associated raw forecasts from the analogue, hydrological and meteorological models (produced by the UK Met Office, Centre for Ecology and Hydrology, British Geological Survey, Environment Agency, Natural Resources Wales, Scottish Environment Protection Agency and Rivers Agency Northern Ireland) were the main sources of SHF information currently being used..."

(1) Line 29: I would be curious to know how/why the users think SHF has potential to forecast floods. I can understand that it may indicate if the season will be drier or wetter than normal, but this is not necessarily an indication that a flood event will happen and action should be taken now, for instance. This is, at least, not a common way of using SHF information.

(2/3) This was indeed interesting! We have edited this paragraph to make it clear that we (the research team) did not give any prior definitions or any guidance on what participants should write down. As such, what was recorded is an accurate representation of how the water sector are currently using SHF and that they do recognise that there is the potential for flood and drought risk to be identified:

"It's important to note that no prior definitions or information were provided and no restrictions or guidance was placed on what participants should write down. This suggests that many in the water sector are using SHF to obtain an insight about whether the upcoming season will be drier or wetter than normal, but that they also believe SHF potentially have the capability to forecast possible flood and drought risk..."

(1) Line 31: what is a "coarse" spatio-temporal resolution to them?

(2) In this case (Hydrological Outlook UK); forecasts that are presented at the scale of whole river basins and updated monthly.

(3) This is now included in the text (page 7, line 6).

(1) Page 6: - Line 29: I think it is a bit disturbing not to show units. I guess the
hydrologists have a feeling that increasing from 1 m3/s to 2 m3/s, for instance, may not be as impacting as increasing from 1 m3/s to 30 m3/s. Graphs without units may be tricky and convey "distorted" information: we have the impression of seeing an important increase in flows when, in fact, it does not represent much, or vice-versa.

(2) This was a point that we discussed at length with our Environment Agency activity co-developers. The concern was that participants who work day-to-day in specific catchments would be very familiar with 'average' and 'high' streamflow and groundwater level records. For example, streamflow values for the Lower Thames at Kingston exceeded 500 m3/s (average flow is 66 m3/s) during the 2013/2014 floods – this was the first time since 1974 that this occurred and is well-cited as an indicator of the extreme nature of the floods. By including the units, some participants would deduce that the SHF must represent the 2013/14 floods, which would bias their decision-making based on their previous experience and memories.

To account for this, we added on the average (Q50) and high-flow (Q10) exceedance thresholds (obtained from 20 years of daily observations) to put the forecasts into context as the weeks and months progressed.

(3) We have added this justification in on Page 9, lines 7-10 to make it clear to the reader.

(1) Have you had reactions in that sense from the participants? How could they place their forecasts with respect to climatology, current hydrological situation, background info, etc. without the "units"? Were these pieces of information also given without units? (see also page 9, lines 4-5)

(2) Yes all information was given without units or dates (sect. 3.3 updated to reflect this). Discussion with the participants suggested that they did not find this concerning or difficult to work with. Based on the Hydrological Summary provided, they deduced that the SHF represented the winter season which we confirmed as this had a key role to play in terms of aquifer recharge, expected rainfall variability and their decision-
making.

(1) - Line 30: if no information on forecast quality was provided, how could the users know that Stage 3 sharper forecast were more skilful (how could they evaluate if the forecasts from all stages were, first of all, reliable?)

(2/3) This is true, they could not assess forecast skill. We have changed use of the word 'skilful' to 'more confident forecasts' in the manuscript as this better represents the activity set-up. With respect to forecast reliability, we have edited the discussion section (Page 21, lines 10-18) to include discussions we had with the participants regarding their evaluations of reliability and how this influenced their decision-making.

(1) - Line 31: why the information was not given for 3 month lead time in all situations? Do you think that the fact of having two systems providing longer lead time information may have influenced the appreciation users might have of the quality of the systems?

(2) Hydrological Outlook UK is only produced as a 3 month outlook, whereas EFAS has potential to forecast out to 7 months (Page 12, line 30). The flood event we wanted to capture was well represented over a 4 month period and we agreed that reducing the SHF all to 3 months was not ideal (as their operational potential to look beyond this). We do agree however, that this may have influenced user's perception and appreciation of the systems and that subsequent discussion about this would have been valuable.

(3) We have included this limitation on Page 24, lines 5-7.

(1) Pages 8-9: - A traditional empathy map is usually used to describe what a type of user think & feel, hear, see, say & do, and also gather their typical "fears" and "gains" in order to create a shared understanding of user's needs. I did not understand well how it was employed here: what were the objectives and how did you build it based on the concepts of the traditional approach? How did you move from the "individual" to the "collective"?

(2) Thank you for this point. The aim of the empathy map was to capture the thoughtprocesses, influences, discussions and the potential risks and gains associated with decisions made on both an individual and on a collective basis.

You are correct, we did adapt the traditional use by asking participants to self-reflect on their decision-making process based on their experiences, knowledge, but also discussions with other group members (thus introducing a collaborative element). By combining (comparing) the information recorded on empathy maps for each group, we also gathered an overview of the shared understanding between forecasters, groundwater hydrologists and water-resource managers. This allowed us to view 'collective' perspectives on how SHF needs and expectations match and differ when it comes to decision-making across the water sector.

(3) We have outlined this more clearly on Page 9, lines 23-31.

(1) From Fig. 4, I have the impression that there are two aspects being "surveyed" in the map: the SHF and the decision. How have you separated these in the analysis?

(2) The maps (Fig. 8) record the SHF decision dots placed on the map. The accompanying participant responses, key statements, quotes and text provided in the results section (Sect. 4.2) are taken from verbal discussions recorded and information written on the empathy maps – this is a qualitative analysis that supports the maps.

(1) Finally, was the concept behind empathy map really needed? I have the impression that the map used here differs significantly (in its target) from the traditional one, which is mostly employed to establish a common ground among team members and to understand and prioritize user's needs. If I understood correctly, basically, a traditional empathy map is built by a group of individuals to reflect a certain type of user, while here it is built by individuals to reflect their own individual preferences, which are them put together to represent a group (although it is not said how this analysis is done in the paper).

(2) Yes, the concept did differ from the traditional use, however, it represented a quick

and effective way of capturing thought-processes, fears, gains, experiences and knowledge at an individual and collective level. From this, we have been able to gain insight and better understanding of the needs of the water sector as a whole.

(3) We have outlined how we employed the empathy maps more clearly on Page 9 and hope the reviewer is happy with this.

(1) I also could not understand how "discussions were captured on the empathy maps" (as stated in line 4, page 20). Could you comment on these issues?

(2/3) We have rephrased this sentence to say: "Discussions captured during the focus group and indicated on some empathy maps identified two key communication barriers in the West Thames"

(1) - While the situation of high flow or low flow was not informed in the beginning (as mentioned by the authors on page 6, line 28), info on flood risk only was provided (page 9, line 13). Isn't there a chance that this has influenced the participants to consider the forecasts as that of an upcoming flood event?

(2) Thank you for raising this very valid point that we had not thought of. While an alternative map for drought risk would be trickier to create (as the impacts are wider reaching and more diffuse in nature), we agree that it is possible that providing a flood map may have influenced the participants.

(3) We have discussed this limitation at the top of Page 24 and highlight that discussions with the participants at the end of the activity with respect to this point would have been useful.

(1) Also, maps were provided showing exceedances of given Q thresholds, which is more common in floods than drought visualization maps.

(2) As these maps accompanied and visualised information from the hydrographs (Stages 2 and 3), it had already become clear to the participants that they were engaging with a potential flood risk scenario by this stage so we do not think that this would

have been an issue.

(1) Finally, was it really necessary to hide that the forecasts were for a potential flood situation? Wouldn't they guess it anyway?

(2) We wanted the SHF to be interpreted as if in a real-world scenario where the participants would not know the upcoming situation in advance.

It became clear that the participants were not sure at Stage 1 (Hydrological Outlook UK) what kind of a situation they were looking at:

"General consensus was for normal or above-normal conditions over the next 3 months, however the information was "too vague to be actionable"" (Page 16, line 25).

It was only once they were given the EFAS forecasts (Stage 2) that they started to make some informed decisions. If we told them the SHF represented a potential flood event in advance, then this would likely have raised some different responses at Stage 1.

(1) Page 9, line 17: do you mean "past (or previous) month"?

(2/3) "Previous" – corrected, thanks.

(1) - Page 18, line 13: I think "low skill" and "more dispersive" forecasts are interchangeably used here. The fact that stage 3 forecasts were sharper does not mean they were more skilful, although the perception of the participants seems to be that they were. I think this needs a more clear explanation. Users may think that the sharper, the better, which is not always the case in SHF. From my experience, water managers tend to understand this aspect much better than flood forecasters.

(2/3) Thank you – we have updated the Discussion to explain these findings in more detail as it was a discussion point we had with the participants. Please see Page 21, lines 10 – 18.

(1) Discussion: - Some parts seem to present results and not a discussion. For instance, page 20 presents, for the first time, the results of the empathy maps analysis. I think this should go to the section on results. By mixing these sections (results and discussion) it is unclear what pertains to the stakeholders' points of view and what comes from the authors' views/perceptions. For instance, when mentioning a series of projects, IMPREX, EUPORIAS, Edge, SWICCA, etc., it is not clear if these projects were discussed with the stakeholders and participants of the activity reported in the paper (and how it influenced, or not, the discussions and responses to the activity).

- Page 21, lines 15-16: it is not indicated anywhere in the methodology of the activity where knowledge on "ongoing scientific developments in SHF" was conveyed to participants. This "conclusion" seems to be disconnected with the methodology presented in the paper. -

(2/3) We please refer the reviewer to Results Section 4.2 where the results from the empathy maps are first presented along with the decision maps (Fig. 8).

Thank you for your point regarding stakeholder points of view and author views. We have rephrased parts of the Methods and Discussion to make this clearer. Firstly we have added a section on the aims of the Focus Group at the beginning of the Methods (Sect. 2.1, page 4) and made the full programme available as a supplement:

"These aims [listed above] were delivered through a series of 4 interactive sessions designed to actively engage participants to share their knowledge and experiences of SHF, and short presentations that introduced the main topics surrounding SHF and informed participants about current SHF projects and developments in the scientific research. While this paper focuses on the decision-making activity (interactive session 2), discussions from the other sessions are also presented where relevant. An outline of the focus group programme is provided in Supplement 1…"

We have also rephrased the Discussion section (Page 22, lines 26-28) so that it is clear that participants were introduced to ongoing scientific developments and discussed SHF projects as part of the focus group.

(1) Page 21, lines 19-21: I am a bit annoyed here because of the original concept behind "empathy maps", as mentioned earlier in my remarks. The definition of "empathy", itself, is related to the action of understanding, being aware of, being sensitive to, etc. the "other" (not oneself), while here it is used by the participants to express their own feelings. They use it as a mirror and not as a way to "put yourself in someone else's shoes". My question is: is it really the appropriate tool to address the issues you want to address? Why did the authors choose this specific tool? I believe the authors got the answers to their questionings, but I have the feeling that this was done with a tool that was fully adapted by them to that purpose, and not as a result of applying the well-known technique of the "empathy map". I think this needs to be clarified.

(2) Thank you – we believe we have addressed this point above.

---

## Author Comment (AC3) · 21 Sep 2018

We thank reviewer 2 for taking the time to review our manuscript and for providing helpful comments and insight. We've edited our manuscript accordingly and our responses to all the reviewer's comments are included below. We have used the following sequence: (1) comments from Referee, (2) author's response, (3) author's changes in manuscript.

*General comments* I would like to thank the authors for this innovative contribution that addresses a still little-explored topic in the (seasonal) hydrological forecasting research community. The proposed article explores forecast uncertainty, forecast com-

munication and decision-making with an original experiment involving and giving voices to decision-makers from the West Thames. As highlighted by the authors, this is a research area with limited contributions, which, in my opinion, fits nicely in Geosciences Communication. The paper is didactic and well documented, and I strongly recommend it for publication, though I do have some minor questions which I list below.

\*Specific comments\*

(1) Section 2.2.1: (a) Here, I think some more information on the process of finding participants would be useful for the reader. From this section, it seems that all eleven invited participants agreed to be part of the focus group: "11 West Thames stakeholders [: : :] were invited to take part in the focus group [: : :]" (L.12). If so, I assume there were previous collaborations, and did these have any role in the willingness to participate? And if not, how many stakeholders were invited, how many declined and, if any, for which reasons?

(2/3) Thank you for this point – we have added more information about how many participants were originally invited (17), how we chose to invite participants (which was largely based on previous collaborations or people known to hold established positions relevant to SHF in the West Thames) and how many participants declined (5) with explanations given as to why where this was known (Page 6, lines 8 – 18).

(1) (b) Additionally, it would be interesting that the authors mentioned how many forecasters, public water suppliers, waste water modellers and operators, etc. are active in the area. For example, how representative are the 3 forecasters that took part in the group? What do 11 stakeholders represent at the scale of the region? This is for the sake of giving a wider picture to the reader on the stakeholders being active in the area, and on the impact/outreach this experiment has had.

(2) We're afraid that we don't have this information. However, we did make sure that those who were invited held established positions (i.e. had long-term experience of working in their role and at their organisation thus would be in a good position to represent their viewpoint). We also ensured that the major water sector organisations in the West Thames were represented.

With respect to outreach, Sect. 5.4 details the implications for future policy and decision-making from the Environment Agency's perspective which was raised directly as a result of our Focus Group.

(1) (c) How many different organizations were represented through these 11 participants?

(2) 5, including Government agencies, public bodies, water utilities companies and not-for-profit organisations.

(3) This is now clarified in Sect 2.3.1.

(1) Section 3.2.2: How much do you think the group opinion influenced the colour chosen by individual participants? Could there be biases here?

(2) There was the potential for bias here, however, discussions would often take place in real-life and we did reiterate to participants that the colour they chose should represent what they or their organisation would do with the information. The responses (Fig. 8) do differ between same-group members which suggests that bias was not a major issue.

(1) Figure 3: It is mentioned in the text that there are about 110 dots on each map (L.9), but in Figure 3, we observe around 9 or 10 dots per catchments. What is the reason for this? Were some stakeholders only working on their usual catchments of interest?

(2) Yes, some participants said that they only felt comfortable providing decisions for catchments they were familiar with. In some cases, participants also felt that they were not able to make an informed decision and so did not place a dot.

(3) We have clarified this more clearly on Page 9, lines 22-23.

(1) Section 3.3.1: It was not clear to me how familiar the participants were to this
information and what they got out from it if they are already familiar with the region.

(2) Some participants were very familiar with the West Thames as a whole and the hydrological risks and opportunities that the different catchments present, whilst others were less so i.e., where they work on just a few catchments on a day-to-day basis. The background maps were used to kick-off discussions and to make sure that everyone was familiar with the wider area being used for the activity (now clarified in Sect. 3.3.1). From our perspective, the maps also allowed us to find out what factors different users of the water sector focus on (Sect. 4.4.1).

(1) Section 4.2: "At no point did participants ignore the SHF information" (L.18) Isn't this result due to the context? From the moment the participants know they are in a seasonal forecasting experiment, they are willing to use the provided information.

(2) We believe not – we reiterated to all participants that we would like their decisions to be representative of what they or their organisation would do with the SHF information in real life. As we had already identified that participants are engaging with SHF to some degree (Sect. 2.3.2), we knew that they would likely be willing to use the information. If we had found out that the water sector were not currently using SHF, then the activity would have taken on a more hypothetical focus.

(1) Section 5.3 (L.22-29): How did the authors deal with forecast quality in this experiment? (a) From this paragraph, it seems that no quality information was provided, and indeed, no quality information appears in the Stage 1, Stage 2 and Stage 3 sections of the Supplement. Was it a choice to exclude this information, or is it not available to users in the Hydrological Outlook UK and the EFAS-Seasonal?

(2/3) You are correct, we did not provide forecast quality information (we have clarified this on Page 9, line 10) for 2 main reasons. Firstly, quality information is not routinely provided in Hydrological Outlook UK and EFAS-Seasonal. Secondly, during the co-development of the activity, the Environment Agency recommended that introducing information on SHF quality / skill may make the activity too complex, especially if there

were participants who were not familiar with using SHF at all. We did however discuss the importance of forecast quality with participants and present this in the Discussion on Page 21, lines 4 – 18.

(1) (b) In the absence of quality information, did the users assume that forecast quality was the current/latest one they are aware of?

(2) Yes – participants were broadly familiar with Hydrological Outlook UK and so during discussions they said they interpreted the information in the same way as they would in real-life. By contrast, at Stage 3 ('Improved' EFAS-Seasonal) the quality of the forecast was questioned because it was beyond what is expected by current SHF.

(1) (c) Several platforms now propose quality information along with the forecasts, and assuming that this information is provided in a clear manner, users do not have to hypothesize about the quality of the forecasts they use. In this specific paragraph, authors suggest "to keep water sector users informed of scientific developments". In my opinion, it is also crucial to provide quality information in an intelligible way along with the forecasts, as well as build the required user knowledge to understand this information.

(2/3) Thank you, this is important and we have added this point on Page 23, line 4.

*Technical corrections*

(1) Figure 8: A reminder of the colour codes would be useful for the reader to have.

(2/3) Thank you, yes we agree – we have added this information to the legend for Figure 8.

---

## Author Response (AR2)

Dear Katharine Welsh,

Thank you for taking the time to review our responses to the referees. We have uploaded our revised manuscript (and additional supplement) for your consideration. We have also addressed additional points that you raised in our manuscript and have commented on these below.

We look forward to publishing our article in *Geoscience Communications*.

Best wishes
Jess Neumann et al.

Dear Jess et al.,

Thank you for your timely revisions to the manuscript which now reads very clearly and benefits from the addition of the participant table.

Having read the manuscript in full with revisions, if you have the data available, I think section 5 as a whole would benefit from some quotes from participants to illustrate the points that you make. Including but not limited to page 23 line 23 where you describe the participants "err on the side of caution", the use of a quotation here would illustrate the point more clearly than a summary in your own words. Similarly page 24 line 19 "many participants said…" could be illustrated using a direct quotation from your data.

Thanks – this is a good idea. We have added some quotes in section 5 to help illustrate these points more clearly. In some cases (e.g. page 24, line 19) the points were made through general discussion and so we don't have data for direct quotes, but where we do, we have incorporated these.

Some other minor technical points :

Page 4 Line 11 – perhaps 'subsequent' would be preferable to 'knock-on' as GC has an international audience

We have rephrased this to say 'subsequent knock-on' effects to help make this clearer for an international reader. We would like to retain the words 'knock-on effects' in the sentence as it is defined as when an event or situation (e.g. flood) causes other events or situations, but not necessarily directly.

Page 9 Line 5 should now say Table 2

Thank you - corrected

Page 21 Figure 8 caption should now say Table 2

Thank you - corrected

Please consider correcting the following word contractions:

Page 8 Line 17 and Page 24 Line 1 (it's --> it is)

Page 25 Line 2 (don't --> do not)

Page 25 Line 3 (haven't --> have not)

Thank you – we have corrected these at all uses in the text.

Dear Jessica et al.,

Thank you for your submission to Geoscience Communication and thank you for your clearly labelled responses to the reviewer's comments. Based on your detailed responses to the reviewer's comments, I am recommending that the paper is published subject to minor revisions. I would encourage you to upload a revised version of the article to Geoscience Communication so that an editor can review the article to ensure that the revisions have been made and all fit suitably within the context of the revised article. At present there is no revised version of the paper uploaded to the interactive discussion and whilst I feel satisfied that your responses are detailed, I would prefer to see the article in its entirety as there have evidently been considerable alterations made.

Additional points

1. Whilst I appreciate you may not have the data available to address the question raised by reviewer 2 (1b – how representative are the participants), it could be useful to give a better sense of who the participants were. You state that participants represented Government agencies, public bodies, water utilities companies and not-for-profit organisations; which of the job titles link to which organisation? For example, is the waste water modeller from a public body or a not-for-profit. Similarly, your response to the reviewers comments that the participants who were selected "had long-term experience"; if you have the information available about approximate level of experience of each participant, it might also be useful to include that information. A simple table could include this information and would give the non-hydrology audience of Geoscience Communication a better sense of who the participants represent.

Absolutely – we have included a Table (page 6) which details this information for the reader.

2. It would be useful if you could include the rationale against potential bias that you allude to in your comments to reviewer 2 within the article. E.g. "There was the potential for bias here, however, discussions would often take place in real-life and we did reiterate to participants that the colour they chose should represent what they or their organisation would do with the information. The responses (Fig. 8) do differ between same-group members which suggests that bias was not a major issue."

Thank you for this – we have made this point clearer in Sect 3.2.2 and included a discussion point in the Sect 5.5 Learning Outcomes (page 25, lines 8-10) that highlights the potential for this bias which would benefit from further discussion:

[revised manuscript text omitted]